# On Uniformly Scaling Flows: A Density-Aligned Approach to Deep One-Class Classification

## Abstract

Unsupervised anomaly detection is often framed around two widely studied paradigms. Deep one-class classification, exemplified by Deep SVDD, learns compact latent representations of normality, while density estimators realized by normalizing flows directly model the likelihood of nominal data. In this work, we show that uniformly scaling flows (USFs), normalizing flows with a constant Jacobian determinant, precisely connect these approaches. Specifically, we prove how training a USF via maximum-likelihood reduces to a Deep SVDD objective with a unique regularization that inherently prevents representational collapse. This theoretical bridge implies that USFs inherit both the density faithfulness of flows and the distance-based reasoning of one-class methods. We further demonstrate that USFs induce a tighter alignment between negative log-likelihood and latent norm than either Deep SVDD or non-USFs, and how recent hybrid approaches combining one-class objectives with VAEs can be naturally extended to USFs. Consequently, we advocate using USFs as a drop-in replacement for non-USFs in modern anomaly detection architectures. Empirically, this substitution yields consistent performance gains and substantially improved training stability across multiple benchmarks and model backbones for both tabular and image anomaly detection. These results unify two major anomaly detection paradigms, advancing both theoretical understanding and practical performance.

## 1 Introduction

Unsupervised anomaly detection seeks to identify rare deviations from normality without access to labeled outliers and underpins safety-critical applications in medical diagnostics, cybersecurity, and industrial inspection Fernando et al. (2021); Liu et al. (2024). For such complex underlying data, performance and reliability hinge on how well "normal" structure is captured in a latent space, where typical samples form a compact, semantically meaningful region, so that unlikely patterns can be identified. Estimating the data's true underlying density provides a principled foundation to accomplish this task Ruff et al. (2021).

Within the landscape of leading deep unsupervised anomaly detection methods, there exist two seemingly complementary lines: one-class classification — epitomized by Deep SVDD Ruff et al. (2018) — which forgoes explicit density mapping and learns encoders that concentrate normal data into a minimal-volume support (e.g., a hypersphere), and distribution maps, which estimate a distribution over embeddings Liu et al. (2024). Among the latter, a predominant example are normalizing flows, generative density estimator models which learn bijective transformations between the complex data space and simple base distributions and enable exact likelihoods Papamakarios et al. (2021). Figure 1 schematically contrasts hyperspherical one-class mapping against a normalizing flow with isotopic gaussian base distribution. While one-class methods offer geometric simplicity and efficiency, the risk trivial solutions, hypersphere collapse, and limited density fidelity Chong et al. (2020); Zhang & Deng (2021). Flows provide explicit likelihoods but their scores can be confounded by input-dependent log-determinant terms and higher computational cost. Still, despite different mechanisms, both lines aim to delineate a high-mass region of normality, while a rigorous account of when and how exactly their objectives coincide and how to leverage this in practice has been missing.

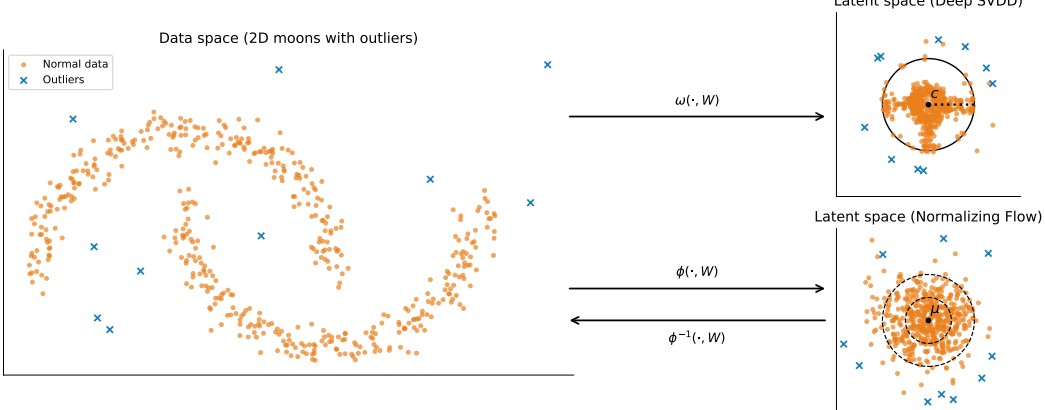

Figure 1: Schematic comparison of Deep SVDD and Normalizing Flows. The left panel shows the input data space: a nonlinearly structured 2D "moons" dataset with outliers. Top-right depicts Deep SVDD's latent space: a learned mapping $\omega(\cdot, W)$ concentrates normal samples near a center $c$ inside a hypersphere, while anomalies lie outside; the non-uniform distribution emphasizes representation bias. Bottom-right shows the Normalizing Flows latent space: an invertible mapping $\phi(\cdot, W)$ moves data towards a standard Gaussian (illustrated by $1\sigma$ and $2\sigma$ contours) and relegates outliers to low-density regions, with $\phi^{-1}(\cdot, W)$ indicating the reverse mapping.

We close this gap by introducing a theoretically grounded bridge via uniformly scaling flows (USFs) — normalizing flows with a constant Jacobian determinant — by showing that USFs are a special case of a Deep SVDDs. In particular, we formally establish that USFs trained via standard maximum likelihood estimation (MLE) are mathematically equivalent to Deep SVDD objectives with a distinctive regularization scheme. Since this regularization directly relates to the determinant of the Jacobian matrix, it penalizes volume collapse in a way that provably circumvents the well-known degeneracy issues of standard deep one-class classification. Motivated by these properties, we advocate USFs as drop-in replacements for state-of-the-art, non-USF-based anomaly detectors. Eliminating input-dependent log-determinant terms makes scores depend only on position in the base latent, yielding more predictable level-set geometry and empirically showing consistent gains across standard industrial AD benchmarks in both image- and pixel-level detection with improved performance stability.

The remainder of this paper is structured as follows. §2 reviews the necessary background on anomaly detection, Deep SVDD, and normalizing flows, formally defining uniformly scaling flows. §3 situates our work within the broader landscape of related research on deep one-class classification and flow-based anomaly detection. Our core theoretical contribution is presented in §4, where we prove the equivalence between maximum likelihood training of a USF and a regularized Deep SVDD objective, formally bridging the two paradigms. Additionally, we analyze the advantages of USFs with respect to common Deep SVDD and non-USF failure modes theoretically and empirically. Motivated by these theoretical advantages, §5 presents an extensive empirical study, where we compare USF and non-USF architectures for tabular anomaly detection and replace standard flows with USFs in modern image anomaly detection architectures (FastFlow, CFlow, U-Flow), showing the potential for consistent performance gains and significantly improved stability across established benchmarks. Finally, §6 concludes.

## 2 PRELIMINARIES

To first define unsupervised anomaly detection formally, let $X$ be a random variable governed by a probability density function $p_X(x)$ that models the true distribution of normal data. An observation $x$ is deemed anomalous if its likelihood under this distribution falls below a predefined threshold $\tau$, i.e.,

$$\{\, x \mid p_X(x) < \tau \,\}. \tag{1}$$

This framing characterizes anomalies as those points residing in the low–probability tails of the normal data distribution, consistent with the definition that "an anomaly is an observation that deviates considerably from some concept of normality" Ruff et al. (2021). One can transform this likelihood criterion into an anomaly score by considering the Shannon information, $-\ln p_X(x)$ (basis $e$, nats).

**Deep SVDD**   Rather than estimating the true density $p_X(x)$ and applying a threshold test as in Equation 1, Deep Support Vector Data Description (Deep SVDD) Ruff et al. (2018) casts anomaly detection as a one-class classification problem: it learns a mapping s.t. the latent representations of normal samples are confined within a minimal-radius hypersphere around a fixed center point.

To this end, Deep SVDD parameterizes a deep neural network $\omega(x, w)\colon X \to \mathbb{R}^d$ with layer weights $w = (w^1, \ldots, w^L)$ and learns these weights by optimizing the one-class loss

$$\min_w \ \frac{1}{n}\sum_{i=1}^{n}\|\omega(x_i, w) - c\|^2 \ + \ \frac{\lambda}{2}\sum_{l=1}^{L}\|w^l\|_F^2 \tag{2}$$

where $\{x_i\}_{i=1}^n$ are the normal training samples and $c \in \mathbb{R}^d$ is the hypersphere center. The second term corresponds to a weight decay regularizer with hyperparameter $\lambda > 0$ where $\|\cdot\|_F$ denotes the Frobenius norm. To avoid the trivial collapse solution of mapping every data point to the center $c$, usually $c$ is fixed to a vector $\neq 0$, bias terms are omitted and only unbounded activation functions are used. In practice, the network weights $w$ are initialized by pretraining an autoencoder on the normal data and adopting its encoder parameters $w_{\text{init}}$ for $\omega$, after which the center $c$ is set to the mean of $\{\omega(x_i, w_{\text{init}})\}_{i=1}^n$. At inference, the squared distance $\|\omega(x, w) - c\|^2$ between an input's latent representation and the hypersphere center is treated as the anomaly measure.

**Normalizing Flows**   Flow models are commonly used as density estimators and generative models. Formally, a flow $\phi$ transforms a usually complex data (or target) distribution $X$ into a typically simple base (or latent) distribution $B = \phi(X)$ using a continuous invertible map with continuous inverse, i.e. a diffeomorphism. The map $\phi_w$ is implemented by an invertible neural network with parameters $w$. We obtain a density estimator and a generative model by applying the flow in both directions:

1. Sampling is performed by first sampling $z \sim B$ and then computing the inverse map $\phi_w^{-1}(z)$.
2. The likelihood $p_w(X = x)$ is computable with the change of variables formula Folland (1999):

$$p_w(X = x) = |\det J_{\phi_w}(x)|\, p_B(\phi_w(x)), \tag{3}$$

where $J_{\phi_w}(x) = \frac{\partial \phi_w}{\partial x}$ denotes the Jacobi matrix of $\phi_w$. Flows are typically trained to estimate the density of some unknown distribution $X$ by fixing the base distribution and maximizing the log-likelihood of samples from $X$ using Equation 3. In this case, we write $\hat{p}_X^w(x)$ in order to avoid confusion with the true density $p_X$. The by far most common base distribution is a standard Normal $\mathcal{N}(0, I)$, which motivates the name normalizing flow.

While most neural networks architectures are intrinsically differentiable, they typically do not enforce bijectivity. One needs to design specific architectures that restrict the hypothesis space to diffeomorphisms. Most discrete (non-ODE) architectures rely on various types of coupling layers Dinh et al. (2015; 2017); Kingma & Dhariwal (2018); Kohler & Roth (2023). Another crucial building block are bijective affine transforms Hoogeboom et al. (2019), which are often restricted to some subclass like (soft) permutations or householder reflections Dinh et al. (2015); Tomczak & Welling (2016).

**Uniformly Scaling Flows**   A core contribution of this work is to reveal a profound theoretical connection between the prominent Deep SVDD anomaly detection framework and a well-known subclass of normalizing flows. A **uniformly scaling flow (USF)** is a diffeomorphism $\phi : \mathbb{R}^d \to \mathbb{R}^d$ whose Jacobian determinant is constant throughout the input space:

$$\det J_\phi(x) = \kappa \quad \forall \mathbf{x} \in \mathbb{R}^d \tag{4}$$

for some constant $\kappa \in \mathbb{R} \setminus \{0\}$. This property induces uniform volume scaling across all regions of the latent space. Note that if $\phi = \phi_w$ is parameterized, then $|\det J_{\phi_w}(x)| = \psi_{\text{det}}^\phi(w)$ for some function $\psi_{\text{det}}^\phi$ that depends only on the parameters $w$.

While non-uniformly scaling flow architectures (e.g., RealNVP) have largely superseded uniformly scaling variants like NICE in anomaly detection architectures, recent theoretical insights demonstrate that USFs retain unique advantages in specialized domains including neuro-symbolic verification Abu Zaid et al. (2024). The USFlows library provides modern and robust implementation of USFs via additive coupling layers, adjoint bijective affine group actions parameterized by learnable general LU-decomposed bijective affine transformations, and invertible $1 \times 1$ convolutions. We will use the USFlows architecture for all our flow experiments in Section 4. In order to provide fair baseline comparisons, we also implemented a non-US counterpart, NonUSFlows, that is identical to the USFlows architecture, except that additive coupling is replaced by affine coupling, which is non-US Dinh et al. (2017). A full description of the architectures is included in Appendix B.1.

## 3 RELATED WORK

**Modern Deep One-class Classification**   Deep one-class approaches such as Deep SVDD are vulnerable to representational collapse (forcing diverse normals into a single center), and latent misalignment (no incentive to preserve input-space density/manifold structure). To make collapse suboptimal at the objective level, follow-up methods inject training signals that enforce feature diversity and margin: DROCC synthesizes near-manifold "virtual anomalies" and learns to separate them from normals Goyal et al. (2020); DeepSAD adds a pull–push term using few labeled outliers to tighten the normal region Ruff et al. (2020); and  Chong et al. (2020) regularize against constant embeddings via noise injection and a mini-batch variance penalty. In parallel, a complementary line of work preserves latent fidelity by coupling compactness with reconstruction or probabilistic consistency: DASVDD and DSPSVDD combine Deep SVDD with autoencoder reconstruction to keep features informative and curb collapse Zhang & Deng (2021); Hojjati & Armanfard (2024), while Deep SVDD-VAE introduces a probabilistic latent that regularizes geometry and improves distributional alignment Zhou et al. (2021). Most recently, Zhang et al. (2024) proposed DOHSC, introducing an orthogonal projection layer into Deep SVDD, incentivizing the latent mapping to be more closely aligned with a spherical shape. Their extension DO2HSC expands up on this approach with a bi-hypersphere compression objective. In contrast to such extensions and auxiliary tasks, we show how training a USF via MLE is equivalent to optimizing a one-class loss with implicit density regularization and how this avoids representational collapse and benefits latents space alignment.

**Normalizing Flows for Anomaly Detection**   Experience with pixel-space flows revealed that maximum-likelihood training steers toward generic low-level regularities. In term, training flows on deep feature embeddings resolves this behavior and markedly improves performance Kirichenko et al. (2020). DifferNet is the first method that fits a flow on features from a pretrained CNN to demonstrate strong image-level detection on industrial AD Rudolph et al. (2021), while CS-Flow extends this line with multi-level/contextualized feature modeling Rudolph et al. (2022). CFlow conditions the flow on spatial indices and backbone features to model the distribution of normal patches per location, enabling accurate localization Gudovskiy et al. (2022); FastFlow implements a lightweight 2D convolutional flow directly on feature maps, preserving topology end-to-end for real-time anomaly segmentation Yu et al. (2021); and U-Flow adopts a U-shaped, multi-scale conditioning design that couples local likelihoods with broader context Tailanián et al. (2024). All of these methods employ variable-volume flows with input-dependent Jacobians. This boosts per-block modeling power, but can confound scoring by entangling base-density and volume terms. Notably, uniformly scaling flows may serve as direct drop-in replacements without having to adjust other methodological components.

**Intersection of Normalizing Flows and Deep One-Class Classification**   Normalizing flows and deep one-class classification both define "normality" as a high-probability (or minimum-volume) region and flag points outside it. Density thresholding with flows and distance cut-offs with one-class objectives are thus two lenses to identify the boundary of normal data. A growing body of research also proposes hybrid approaches. Flow-based SVDD instantiates a SVDD objective atop a volume preserving flow, avoiding collapse while learning a tight normal region in latent space Sendera et al. (2021). One-Flow similarly uses a flow to find a minimal-volume hypersphere that contains a target fraction of the data, operationalizing one-class support estimation Maziarka et al. (2022), while NFAD trains a flow on normal data and samples low-density regions to synthesize pseudo-anomalies that refine one-class classifiers, turning their optimization into a self-supervised binary task Ryzhikov et al. (2021).

**Novelty Relative to Prior USF Approaches** While these hybrid approaches demonstrate the potential of combining flows with one-class principles, they rely on custom objectives that depart from standard maximum likelihood training. Our work reveals that a more fundamental connection exists: *standard MLE training of USFs* itself implements a well-regularized Deep SVDD variant. This precise theoretical bridge—showing the objective-level equivalence between these seemingly distinct paradigms—has been missing from the literature. Our subsequent analysis therefore addresses: when does maximum likelihood training naturally replicate one-class approaches, and how can this intrinsic connection be leveraged to improve performance and robustness in flow-based anomaly detection?

## 4 COMPARING DEEP SVDDS AND UNIFORMLY SCALING FLOWS

This section first establishes a theoretical bridge between Deep SVDD and uniformly scaling flows. It then relates their shared failure modes, delineates native differences in latent space alignment, and shows how dimensionality reduction methods established for Deep SVDD carry over to USFs. Details omitted from this section are contained in Appendix B.

### 4.1 LOSS FUNCTION

Recall that the training loss of a Deep SVDD given in Equation 2 consist of two terms: the square distance from the center $\|\omega(x, w) - c\|^2$ and a regularization term. Comparing the Deep SVDD objective to the maximum likelihood training objective of normalizing flows reveals a close relationship. Specifically, when training a uniformly scaling flow $\phi$ with base distribution $\mathcal{N}\left(c, \frac{1}{2}I\right)$ and parameters $w$, the objective becomes:

$$
\begin{aligned}
\min_w \mathbb{E}_{x \sim X}\left[-\ln \hat{p}_X^w(x)\right] &= \min_w \mathbb{E}_{x \sim D}\left[-\ln \mathcal{N}\left(\phi_w(x); c, \frac{1}{2}I\right)|\det J_{\phi_w}(x)|\right] \\
&= \min_w \mathbb{E}_{x \sim D}\left[-\ln \exp\left(-\frac{(\phi_w(x) - c)^T 2I(\phi_w(x) - c)}{2}\right) - \ln \psi_{\det}^\phi(w)\right] \\
&= \min_w \mathbb{E}_{x \sim D}\left[(\phi_w(x) - c)^T(\phi_w(x) - c) - \ln \psi_{\det}^\phi(w)\right] \\
&= \min_w \mathbb{E}_{x \sim D}\left[\|\phi_w(x) - c\|^2\right] - \ln \psi_{\det}^\phi(w) \qquad (5)
\end{aligned}
$$

Thus, the objectives differ primarily in the "regularization" term that depends only to the weights. We will see in Section 4.2 and Section 4.3 that this alternate term is critical to avoid hypersphere collapse and to ensure a meaningful latent space alignment.

### 4.2 HYPERSPHERE COLLAPSE AND EXPLODING DETERMINANTS

Deep SVDD's objective admits a degenerate solution: *hypersphere collapse*, where all inputs map to a single point, minimizing the loss trivially. This is mitigated architecturally (e.g., no biases, fixed center) but limits expressivity.

Flow-based models circumvent this degeneracy theoretically. Maximum likelihood training of a flow minimizes the reverse Kullback-Leibler divergence,

$$
\begin{aligned}
\text{KL}\left(P_{\phi(X)} \,\|\, \mathcal{N}\left(c, \frac{1}{2}I\right)\right) &= \mathbb{E}_{x \sim P_X}\left[\ln \frac{p_X(x) \cdot |\det J_\phi(x)|^{-1}}{\mathcal{N}\left(\phi(x); c, \frac{1}{2}I\right)}\right] \\
&= \underbrace{\mathbb{E}_{x \sim P_X}[\ln p_X(x)]}_{\text{const.}} - \mathbb{E}_{x \sim P_X}\left[\ln \mathcal{N}\left(\phi(x); c, \frac{1}{2}I\right) + \ln \psi_{\det}^\phi\right] \\
&= \mathbb{E}_{x \sim P_X}[-\ln \hat{p}_w(X = x)] + C, \qquad (6)
\end{aligned}
$$

driving the transformed data distribution toward the base distribution $P_B$ (here, $\mathcal{N}(c, \frac{1}{2}I)$ to align with Deep SVDD). This holds for any base distribution and any flow.

However, in practice, normalizing flows are susceptible to a related pathology: *exploding determinants* Behrmann et al.; Lee et al. (2021); Liao & He (2021). Gradient-based optimization can exploit

the change-of-variables formula by inflating $|\det J_\phi|$, making the high-density region of the base distribution represent only a vanishingly small portion of the data support. We mitigate this effect with a Bayesian approach for the USFlow architecture, placing a log-normal prior on the absolute value of the determinant to regularize its scale during maximum a posteriori estimation (cf. Appendix B.2).

### 4.3 LATENT SPACE ALIGNMENT

While kernel-based one-class methods are consistent density level-set estimators Ruff et al. (2018), Deep SVDD foregoes direct density modeling. Its objective learns compact representations but lacks a mechanism to enforce alignment between input-space density and latent-space geometry. This permits pathological solutions, such as *density inversion*, where a less likely input is mapped closer to the center $c$ than a more likely one, violating fundamental anomaly detection principles while achieving minimal loss.

**Proposition 1.** *Let $\mathcal{N}$ be a $d$-dimensional standard normal distribution with $d > 2$. There exists a class of functions $F_\alpha : \mathbb{R}^d \to \mathbb{R}$, $\alpha \in \mathbb{R}^+$, such that $F_\alpha$ is non-degenerate for all $\alpha > 0$ and $\lim_{\alpha \to \infty} L(N, F_\alpha) = 0$, yet for all $\alpha > 0$ and $x, y \in \mathbb{R}^d$ with $x, y \neq 0$,*

$$|F_\alpha(x) - c| < |F_\alpha(y) - c| \implies \mathcal{N}(x) < \mathcal{N}(y),$$

*where $L$ is the Deep SVDD loss (without regularization) with center $c$.*

*Proof Sketch.* Assuming $c = 0$, define $F_\alpha(x) = 1/(\alpha\|x\|)$ with $F_\alpha(0) = 0$. This inverts densities: as $\|x\|$ increases (lower $\mathcal{N}(x)$), $|F_\alpha(x)|$ decreases. The loss is $L(\mathcal{N}, F_\alpha) = \alpha^{-2}\mathbb{E}_{x \sim \mathcal{N}}[\|x\|^{-2}] = 1/(\alpha^2(d-2))$, which vanishes as $\alpha \to \infty$. □

A discussion of the Deep SVDD loss with regularization is included in Appendix B.3.

In contrast, USFs guarantee density-preserving alignment. From Eq. 3, a constant Jacobian determinant implies $p_X(x) < p_X(y) \Leftrightarrow p_B(\phi(x)) < p_B(\phi(y))$. For an isotropic Gaussian base distribution, this yields the desired property: $p_X(x) < p_X(y) \Leftrightarrow \|\phi(x)\| > \|\phi(y)\|$. This ensures the latent norm is a faithful anomaly score, a property not guaranteed by non-uniformly scaling flows.

We demonstrate this practical advantage on an asymmetrical Gaussian mixture toy dataset (designed to incentivize non-uniform volume transfer), comparing the relationship between data density and latent norms for Deep SVDD, USFs, and non-USFs across dimensions $d = 2, 8, 32, 128$. Figure 2 visualizes the latent space alignment for the 2D case. The exact experimental setup and the full evaluations for all dimensionalities regarding true log-likelihood compared to latent norm and estimated log-likelihood are given in Appendix C.

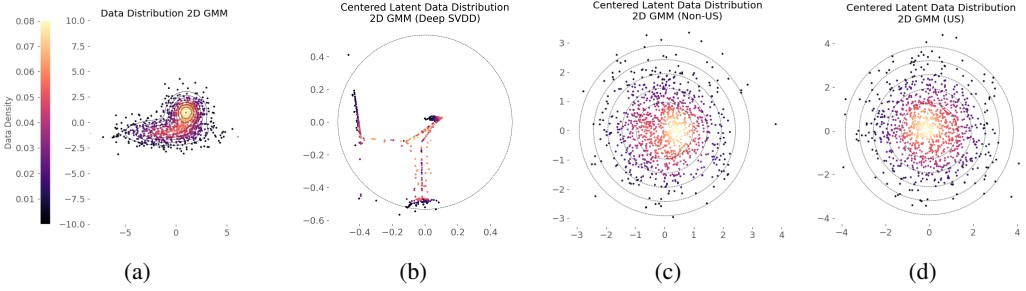

Figure 2: 2D Gaussian mixture experiment. 2a: True data distribution. 2b: Latent space of a **Deep SVDD**. 2c: Latent space of a **non-USF**. 2d: Latent space of a **USF**. Point color encodes the true data density. Dashed lines show countours of the base distribution ($\sigma$–$3\sigma$) (flows) or the decision threshold (Deep SVDD) based on the 99th distance percentile – The USF shows direct density alignment between data and latent spaces, while Deep SVDD and non-USF show noticable discrepancies.

### 4.4 DIMENSIONALITY REDUCTION

A key distinction between Deep SVDD and uniformly scaling flows (USFs) is that the diffeomorphism property of a USF requires its latent space to have the same dimensionality as the input data. However, a USF with a normal base distribution implicitly defines an effective reduction to a one-dimensional score: the latent norm $\phi'_w(x) := \|\phi_w(x) - c\|$ produces an anomaly ranking identical to that of the full likelihood, effectively acting as a Deep SVDD objective with a specialized regularizer (cf. Eq. 5).

To enable explicit dimensionality reduction, we propose a hybrid architecture, VAE-USFlow, which integrates a USF as a learnable prior within a variational autoencoder (VAE) framework. This combines the reconstruction-based objective of a VAE with the density-estimation power of a USF, drawing inspiration from joint training techniques in Deep SVDD Hojjati & Armanfard (2024) and prior learning in VAEs Tomczak & Welling (2018). This approach encourages a meaningful compressed representation. More precisely, our joint model $\hat{p}^{\zeta;\eta}_{X,Z}(x,z) = \hat{p}^{\zeta}_{X|Z}(x \mid z)p^{\eta}_Z(z)$ uses the following generative process:

$$B_{\text{prior}} \sim \mathcal{N}(0, I),$$
$$Z \sim \phi^{-1}_\eta(B_{\text{prior}}),$$
$$X \mid Z \sim \mathcal{N}(\text{dec}_\zeta(z), \sigma^2_{\min}I),$$

with $\sigma_{\min} > 0$ sufficiently small. The variational posterior approximation $\hat{p}^{\eta}_{Z|X}(z \mid x)$ is defined as:

$$Z \mid X \sim \mathcal{N}(\text{enc}^{\mu}_\theta(x), \text{enc}^{\Sigma}_\theta(x))$$

We note that the beneficial properties of the USF apply only to this encoder-produced latent space. To mitigate the known issue of autoencoders potentially mapping out-of-distribution data into high-density regions of the prior Ramakrishna et al. (2022), we combine the latent likelihood with a reconstruction score. To the best of our knowledge, this constitutes a novel application of training a flow-based prior within the VAE paradigm for anomaly detection.

## 5 EXPERIMENTS

### 5.1 EXPERIMENTS ON TABULAR ANOMALY DETECTION

We benchmark USFlow and VAE-USFlow on the classical tabular benchmark from ADBench Han et al. (2022), comparing them to their non-US counterparts, modern Deep SVDD variants (DOHSC and DO2HSC Zhang et al. (2024)), and a recent Flow–SVDD hybrid that also relies on a constant-Jacobian determinant flow (OneFlow Maziarka et al. (2022)). This benchmark comprises 47 real-world tabular datasets with dimensionalities between 3 and 1555, covering a broad spectrum of application domains.

### 5.1.1 EXPERIMENTAL SETTINGS

Because USFlows, NonUSFlows, and OneFlow are structurally similar, we adopt a shared configuration, partly inspired by Maziarka et al. (2022). Concretely, we use 4 coupling blocks whose conditioners each consist of 4 hidden layers with width 256. All models are trained with Adam at a fixed learning rate of $10^{-4}$ for up to 200 epochs of batch size 64, with early stopping after 5 consecutive epochs without a training-loss improvement larger than $10^{-4}$. We use a 70/30 train–test split and remove anomalies from the training set (one-class classification setup). For DOHSC and DO2HSC, we largely follow the tabular architecture from the original paper. The models are first pretrained as autoencoders for 10 epochs, after which the encoder is reused for the subsequent optimization stage. This encoder consists of three linear layers with output dimensions 500, 30, and 30, each without bias, followed by an orthogonal projection. We set the percentile parameter $v = 0.05$, corresponding to the median of the hyperparameter values explored in the original work.

### 5.1.2 EXPERIMENTAL RESULTS

Figure 3 summarizes the AUC-ROC performance across all tabular datasets. In particular, the boxplot in Figure 3a reports the aggregate results. USFlow attains the highest mean AUC-ROC (85.61%),

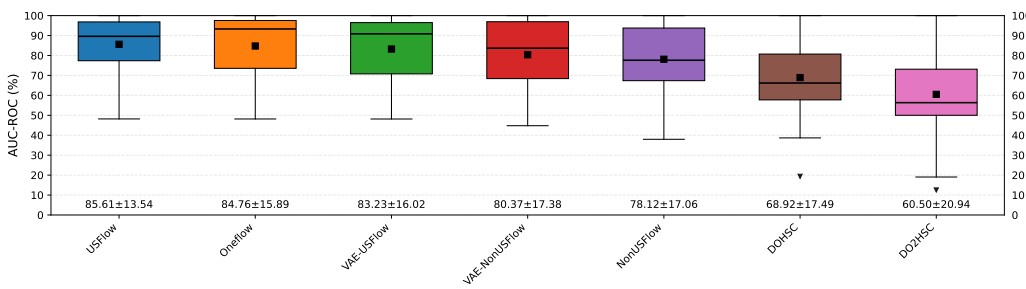

(a) Boxplot aggregations for each algorithm, sorted from left to right by mean. For each box, the line indicates the median, the square the mean and the text below mean ± stdev. performances.

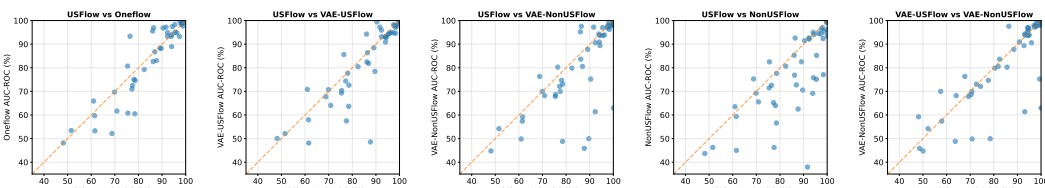

(b) Pairwise comparison of USFLow and USVAEFlow against each other and different baselines.

Figure 3: Overall AUC-ROC performance comparison across all 47 classical ADBench datasets.

while also exhibiting the smallest standard deviation (13.54%) and the narrowest interquartile range. Although its median is slightly lower than that of OneFlow, these results indicate highly competitive performance and markedly increased stability across datasets. VAE-USFlow remains competitive but shows slightly weaker performance. In both cases, the USF variants clearly outperform their non-USF counterparts. The Deep SVDD-based baselines DOHSC and DO2HSC trail all flow-based methods. Figure 3b provides a more fine-grained, pairwise comparison of USFlow and VAE-USFlow on individual datasets, both relative to each other and to the various baselines. When comparing USFlow and OneFlow in particular, we observe that USFlow performs better on the more challenging datasets, where both methods achieve comparatively low AUC-ROC, whereas OneFlow's pairwise wins are primarily driven by even stronger performance on the simpler datasets.

## 5.2 EXPERIMENTS ON IMAGE ANOMALY DETECTION

The objective-level link to Deep SVDD, most notably the spherical alignment of density level sets in the latent space, paired with the observations from our tabular anomaly detection experiments suggest, that USFs may serve as beneficial drop-in replacements within state-of-the art flow-based image anomaly detection. This is due to an important advantage of our theoretical insight compared previously proposed USF based Deep SVDD variants such as OneFlow: Since we leverage maximum likelihood training, our architecture is interchangeable with any other flow model without the need to adopt the loss function. Recent architectures such as FASTFLOW Yu et al. (2021), CFLOW Gudovskiy et al. (2022), and U-FLOW Tailanián et al. (2024) achieve strong performance and are available via the unified ANOMALIB library Akcay et al. (2022), yet they rely on affine coupling layers.

We instantiate a USF layer by (1) replacing affine coupling with *additive* coupling (rendering the flow uniformly scaling) and (2) inserting LU-parameterized bijective affine transforms with fixed diagonal magnitudes to recover per-block expressivity. This US variant is evaluated against its affine (non-US) counterpart across the MVTEC AD Bergmann et al. (2019) and VISA Zou et al. (2022) benchmark datasets using the three aforementioned model architectures.

### 5.2.1 EXPERIMENTAL SETTINGS

To isolate the practical impact of substituting affine (non-US) flows with USFs in unsupervised, out-of-the-box deployment, we adopt the architecture and hyperparameter choices from the original papers and refrain from model- or class-specific tuning. As a consequence, absolute performance values are

lower than leaderboard reports. However, the within-architecture comparisons are strictly controlled and therefore reflect the causal effect of the flow substitution under various conditions, rather than tuning artifacts. The parameterization of the specific architectures is provided in Appendix E.

All experiments were conducted based on Anomalib v2.0.0, using its PyTorch Lightning implementations. Each model is trained for up to 100 epochs with batch size 32 using Adam, with early stopping after three epochs of no improvement. The feature extractor backbones are initialized and frozen with ImageNet-pretrained weights. Every configuration is run three times and we report mean $\pm$ stdev.

### 5.2.2 EXPERIMENTAL RESULTS

| Dataset (Metric) | BASE FLOW | | | USF | | |
|---|---|---|---|---|---|---|
| | FastFlow | U-Flow | CFlow | FastFlow | U-Flow | CFlow |
| MVTec AD (Image) | 91.3±1.5 | 92.7±4.6 | 72.5±12.8 | 91.0±0.9 | **94.5±0.6** | 90.8±1.1 |
| MVTec AD (Pixel) | 95.8±0.4 | 95.9±2.8 | 95.4±1.0 | 96.8±0.2 | **97.0±0.4** | 96.6±0.0 |
| VisA (Image) | 88.0±1.7 | 84.6±6.7 | 82.5±8.3 | 87.7±0.6 | 85.6±0.6 | **88.4±0.9** |
| VisA (Pixel) | 96.3±0.7 | 95.8±3.8 | 97.8±0.5 | **98.8±0.0** | 95.7±1.8 | 98.2±0.0 |

Table 1: Mean AUC-ROC and stdev. (%) over three seeds; dataset values are class-averaged. Highest mean per row in bold, second-highest underlined. Class-specific results are provided in Tables 2- 5.

Across datasets and granularities, replacing the affine base with uniformly scaling flows (USFs) either improves mean performance or leaves it effectively unchanged, while consistently and substantially reducing run-to-run variance. The effect is stable at both image and pixel level.

At a high level, USFs make the strongest impact where base flows are less stable: they markedly tighten variability (often approaching negligible standard deviations) and lift means most for CFlow and U-Flow, while FastFlow typically maintains its mean and still benefits from improved stability. On MVTec AD, USFs yield the most pronounced consistency gains and deliver the largest mean increase for CFlow; on VisA, the best row-wise means shift from U-Flow toward CFlow (image) and FastFlow (pixel) under USFs, with U-Flow remaining competitive and considerably more stable. Fine-granular results for the specific dataset classes are provided in Appendix F. As a proof of concept, we also evaluate VAE–USFlow alongside a version with a NonUSFlow prior on MVTec AD (see Appendix B). While these hybrids do not yet match specialized flow-based anomaly detectors, the USF version achieves a substantially higher average performance and run-to-run performance stability relative to its affine counterpart.

### 5.2.3 ABLATION STUDY

To disentangle which architectural change drives the stability and accuracy gains, we analyze changes against the affine baseline for two uniformly-scaling replacements: (1) an additive USF, which only swaps affine for additive coupling; (2) the full USF from the previous section, which introduces the LU-parameterized general bijective affine transforms as well. Figure 4 reports the deviations in mean and standard deviation per-class on MVTec AD for image-level AUC-ROC (%). The pixel-level counterpart is given in Appendix F). Both ablations yield modest but consistent gains compared to the non-USF version. For U-Flow, for example, the *Additive (USF)* improves mean AUC-ROC by $+2.09$ while reducing the run-to-run standard deviation by $-3.93$, and the *Additive + LU (USF)* achieves a similar improvement of $+1.83$ and $-3.96$ respectively. These findings indicate that the uniformly-scaling (additive) modification is the primary driver of improvements, while the additional LU-based bijective affine transforms can be beneficial in lightweight architectures to preserve expressivity.

For UFlow as the best-performing architecture on MVTec, we also evaluated the effect of integrating the log determinant prior (not directly integrated for minimal changes in the image experiments), as well as changing the backbone. Integrating the log determinant prior showed no significant effect on aggregated performance, neither at the image- ($94.50 \pm 0.40 \rightarrow 94.63 \pm 0.22$) nor at the pixel-level ($96.97 \pm 0.21 \rightarrow 96.55 \pm 0.13$). When switching the standard MS-CaiT backbone to ResNet-18, however, we continue to see a clear advantage of USF over non-USF at image-level ($85.79 \pm 10.70 \rightarrow 84.15 \pm 13.67$), but a slightly weaker pixel-level performance ($94.83 \pm 5.28 \rightarrow 94.95 \pm 3.78$).

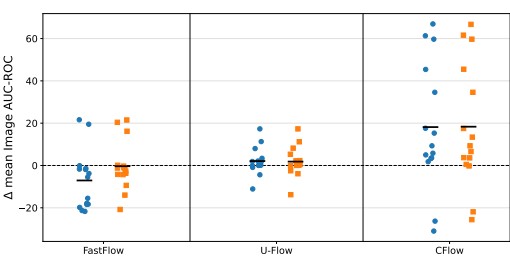 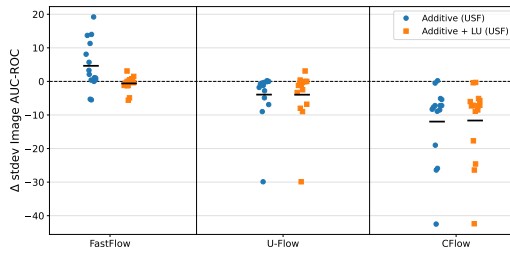

(a) Δ mean image AUC-ROC (higher is better).  (b) Δ stdev image AUC-ROC (lower is better).

Figure 4: Ablation relative to the affine baseline per MVTec AD class. The dashed $y=0$ lines mark the affine baseline; the black horizontal mark is the class-wise mean $\Delta$. *Additive (USF)* swaps affine for additive coupling; *Additive + LU (USF)* additionally inserts LU-parameterized affine transforms.

## 5.3 DISCUSSION

Our experiments show that replacing non-uniformly scaling flows with USF variants consistently improves or preserves detection performance, while systematically reducing run-to-run variance across all architectures. This gain in robustness arises even though the additive coupling layers underlying USFs are strictly less expressive per block than their affine counterparts Draxler et al. (2024), suggesting that removing the input-dependent scaling yields a more favorable inductive bias and numerical behavior than the additional local expressivity afforded by non-uniform scaling.

From an optimization and representation perspective, our experimental findings align with the theoretical advantages of USFs. In non-USFs, the input-dependent log-determinant term in the change-of-variables formula can exhibit high gradients in regions where high- and low-density areas are conflated in the latent space (see Figure 7 in the Appendix). Combined with a weak training signal from low-density areas (the "tails" of the distribution), this can lead to unstable estimation and poor outlier detection in these regions. In contrast, USFs remove the input-dependent volume change, making the anomaly score depend solely on the latent norm (under an isotropic Gaussian base). The diffeomorphism property of the flow, together with the maximum likelihood objective, provides an implicit signal to push low-density areas outward: since high-density areas are aligned closest to the center, the model must allocate less typical instances to the periphery to maximize the likelihood. This results in a more stable and aligned latent space, which in turn improves anomaly detection performance and training stability.

## 6 CONCLUSION

This work establishes a formal equivalence between Deep SVDD and maximum-likelihood training of uniformly scaling flows (USFs), bridging two major paradigms in deep anomaly detection. This theoretical connection reveals that USFs inherently combine the distance-based reasoning of one-class methods with the density faithfulness of flow models, while their constant Jacobian determinant provides a natural regularization against representational collapse. Empirically, we demonstrated that USFs as a simple drop-in replacement for standard flows have the potential to perform highly competitive on tabular anomaly detection and enhance modern image-AD architectures (FastFlow, CFlow, U-Flow), consistently improving stability and often boosting accuracy. These gains align with our theoretical analysis: by eliminating input-dependent volume changes, USFs avoid the confounding log-determinant effects that destabilize tail estimation in non-USFs. The diffeomorphism and MLE objective naturally push low-density areas outward, yielding more stable and aligned latent spaces.

From this perspective, several promising directions emerge. The role of the base distribution warrants deeper investigation, such as employing radial distributions defined by different norms (e.g., L1, Laplacian) or learning the norm distribution $P_{\|B\|}$ itself to better match the latent data structure. Furthermore, exploring more flexible base distributions, such as mixtures or heavy-tailed alternatives, could better capture complex, multi-modal normal data characteristics Abu Zaid et al. (2024); Draxler et al. (2024). Ultimately, this theoretical bridge provides a principled foundation for designing more robust, well-calibrated, and performant anomaly detectors.

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

## A    Use of Large Language Models

We employed large language models (ChatGPT-5, DeepSeek) to assist with editorial polishing of the manuscript (academic tone, clarity, and grammar) and literature discovery and retrieval (suggesting potentially relevant prior work and helping formulate search queries). All technical content was conceived and validated by the authors. Any references surfaced by the model were verified by consulting the original sources, and all citations reflect the authors' own reading. The authors remain fully responsible for the accuracy and integrity of the manuscript.

## B    Formal Proofs and Architectural Specifications

### B.1    Details Omitted from Section 2

In the following, we use $m \odot x$ to denote the component wise (Hadamard) product between two tensors of same topology.

**USFlows Architecture**    According to Abu Zaid et al. (2024), the USFlow architecture is build from masked additive coupling layers Dinh et al. (2015); Ma et al. (2019) and bijective affine transformations parameterized by LU-decomposed matrices Chan et al..

A masked additive coupling layer $C : \mathbb{R}^d \to \mathbb{R}^d$ (or any other fixed tensor topology) defines a bijection based on an arbitrary *conditioner* function $f : \mathbb{R}^d \to \mathbb{R}^d$ and a $\{0, 1\}$-mask $m$ as $C(x) = x + (1 - m) \odot f(m \odot x)$. By construction, $m \odot x = m \odot C(x)$ and hence $C^{-1}(y) = y - (1 - m) \odot f(m \odot y)$.

An LU layer computes a general bijective affine transform $A(x) = Mx + b$. Bijectivity is guaranteed by imposing constraints on the parametrization linear transform: $M = LU$, where $L/U$ are lower/upper triangular matrices with $L_{ii} = 1$ and $U_{ii} \neq 0$ for all indices $i$.

In order to to allow uniform handling of different input topologies, USFlows employs one-star-convolutions (convolutions with kernel size 1 along all spatial dimensions, adapted to input topology) for the bijective affine transforms, where an LU-decomposed channel transform ensures bijectivity. For the conditioner of the coupling transforms, general adaptive convolutions (convolutions with homogeneous kernel size along all spatial dimensions, adapted to input topology) with layer normalization and gating are employed.

A USFlow model is build from blocks of shape $B_i = A_i^{-1} \circ C_i \circ A_i$, where $A_i$ is a bijective affine transform and $C$ is an additive coupling layer. The complete flow is of shape $\phi = A_{n+1} \circ B_n \circ B_{n-1} \circ \cdots \circ B_1$, where the final affine transform ensures that the flow can have an arbitrary (constant) Jacobian determinant. Note that each block is volume preserving since determinants of $A_i$ and $A_i^{-1}$ cancel each other and additive coupling layers are volume preserving by design Dinh et al. (2015).

**NonUSFlows Architecture**    The NonUSFlows architecture is identical except that we use masked affine coupling instead of additive coupling for the blocks.

A masked affine coupling layer $C : \mathbb{R}^d \to \mathbb{R}^d$ (or any other fixed tensor topology) defines a bijection based on an arbitrary additive conditioner function $f : \mathbb{R}^d \to \mathbb{R}^d$, a multiplicative conditioner function $g : \mathbb{R}^d \to \mathbb{R}^d$ with $g(x)_i \neq 0$ for all $x \in \mathbb{R}^d$, $i \in \{1, \ldots, d\}$ and a $\{0, 1\}$-mask $m$ as $C(x) = m \odot x + (1 - m) \odot g(m \odot x)x + (1 - m) \odot f(m \odot x)$. Again, since $m \odot x = m \odot C(x)$, the inverse is given by $C^{-1}(y) = m \odot y + (1 - m) \odot \frac{y - (1 - m) \odot f(m \odot y)}{g(m \odot y)}$. Note that an affine coupling layer with $g(x) = 1$ for all $x$ is equivalent to an additive coupling layer.

Allowing arbitrary non-zero outputs for $g$ can be numerically unstable in practice (locally exploding determinants). Therefore, we adopt the common practice to apply a final scaled $\tanh$ activation with a configurable scaling parameter to our conditioner network $g$. Moreover, $f$ and $g$ are implemented by a single conditioner that computes both outputs, which further unifies the interface of both models. In all our experiments, the parametrization of the conditioner used in USFlows and NonUSFlows differs only in the final layer to produce the required output shapes, respectively.

## B.2 Details Omitted from Section 4.2

**Implicit Determinant Regularization**   We show that it is possible to define a prior on the weights of an USFlow such that the induced prior on the determinant of the transformation is log-normal.

The **bilateral log-normal** is defined on $\mathbb{R}$ as $\mathrm{BiLogNormal}(x; \mu, \sigma^2) = \frac{1}{2}\mathrm{LogNormal}(|x|; \mu, \sigma^2)$, where $\mathrm{LogNormal}(x; \mu, \sigma^2) = \frac{1}{2\sigma\sqrt{2\pi}} \exp -\frac{(\ln x - \mu)^2}{2\sigma^2}$ is the usual log-normal distribution defined on $\mathbb{R}_{\geq 0}$. Note that if $X \sim \mathrm{BiLogNormal}(\mu, \sigma^2)$ then $|X| \sim \mathrm{LogNormal}(\mu, \sigma^2)$ and $\ln|X| \sim \mathcal{N}(\mu, \sigma^2)$.

In the following, we carry out the derivation for image topologies, i.e. input tensors of shape Channels $\times$ Height $\times$ Width. The derivation for other input topologies is completely analogous. Now recall that the USFlows Architecture is build from additive coupling and affine transforms. Additive coupling is volume preserving by design, which means that we only need to take care of the affine transforms. The affine transforms of a USFlow are given by bijective $1 \times 1$ convolutions with LU decomposed bijective channel transforms. Let $C$ be a $1 \times 1$ convolution with channel transform $A = LU$ for lower/upper triangular matrices $L/U$ with $L_{ii} = 1$ and $U_{ii} \neq 0$ for all $i$. For a fixed image topology $c \times h \times w$, $|\det J_C(x)| = |\det A|^{hw}$. Further, since $|\det A| = \prod_{i=1}^{d} |U_{ii}|$, we impose an independent bilateral log-normal prior $\mathrm{BiLogNormal}(0, \sigma_0^2)$ with $\sigma_0 = \sigma/(chw)$ on each $u_i := U_{ii}$. This yields multiplicative regularization:

$$p\left(|\det J_C(x)|\right) = p\left(\prod_{i=1}^{c} |u_i|^{hw}\right) = p\left(\exp\left(hw\sum_{i=1}^{d} \ln|u_{ii}|\right)\right).$$

Since $p(\ln|u_i|) = \mathcal{N}(0, \sigma_0^2)$ and all $u_i$ are independent:

$$hw\sum_{i=1}^{c} \ln|u_i| \sim \mathcal{N}(0, \sigma^2) \implies |\det J_C(x)| \sim \mathrm{LogNormal}(0, \sigma^2).$$

Variance control via $\sigma_0$ thus regularizes determinant scale.

Note that it is sufficient to regularize the affine transforms individually to the desired scale because, except for the last affine transform, all affine transforms appear as adjoint actions $C^{-1} \circ F \circ C$ applied to an additive coupling layer $F$ and therefore determinants cancel block-wise. This prior regularizes the overall transformation as well as the inter-block transforms in a uniform way. Note that we did not observe a need to regularize the remaining parameters of the network in our experiments.

## B.3 Details Omitted from Section 4.3

**Proposition 1**   Let $\mathcal{N}$ be a $d$-dimensional standard normal distribution with $d > 2$. Then there exists a class of functions $F_\alpha : \mathbb{R}^d \to \mathbb{R}$, $\alpha \in \mathbb{R}^+$, such that $F_\alpha$ is non-degenerate for all $\alpha$ but $\lim_{\alpha \to \infty} L(N, F_\alpha) = 0$, and yet for all $x, y \in \mathbb{R}^d$ with $x, y \neq 0$, it holds that

$$|F_\alpha(x) - c| < |F_\alpha(y) - c| \implies \mathcal{N}(x) < \mathcal{N}(y),$$

where $L$ is the Deep SVDD loss (without regularization) with center $c$.

*Proof.* For simplicity, we assume $c = 0$ (otherwise, $F_\alpha$ would also need to shift points towards $c$). We define

$$F_\alpha(x) = \frac{1}{\alpha\|x\|},$$

with $F_\alpha(0) = 0$. Consequently, as $\|x\|$ increases (i.e., as $x$ becomes less probable under $\mathcal{N}$), the center distance $|F_\alpha(x) - c| = |F_\alpha(x)|$ decreases and vice versa.

The overall loss for a function of this class is

$$L(\mathcal{N}, F_\alpha) = E_{x \sim \mathcal{N}}\left[|F_\alpha(x) - c|^2\right] = E_{x \sim \mathcal{N}}\left[\left(\frac{1}{\alpha\|x\|}\right)^2\right] = \frac{1}{\alpha^2} E_{x \sim \mathcal{N}}\left[\frac{1}{\|x\|^2}\right].$$

Since $x$ is a vector in $\mathbb{R}^d$ where each component $x_i$ is an i.i.d standard normal random variable, i.e. $x_i \sim \mathcal{N}(0,1)$, it follows that $\|x\|^2$ is distributed according to a chi-squared distribution with $d$ degrees of freedom, i.e. $\|x\|^2 \sim \chi^2(d)$. Therefore, for $d > 2$, the expected value simplifies to $E_{x \sim N}\left[\frac{1}{\|x\|^2}\right] = \frac{1}{d-2}$. Substituting into the loss gives $L(\mathcal{N}, F_\alpha) = \frac{1}{\alpha^2(d-2)}$, and thus $\lim_{\alpha \to \infty} L(N, F_\alpha) = 0$. $\square$

**Discussion** We choose to ignore the regularization term in our example because we assume such degenerate solutions to be approximated by complex neural networks and the influence of the regularization term w.r.t. the parameterization given in the example can be misleading. For the sake of completeness, however, we would still like to mention that it is not hard to adjust the example such that we can meet arbitrarily small losses with the optimal solution under the Deep SVDD loss function, even if the standard regularization is applied to the given parameters.

### B.4 DETAILS OMITTED FROM SECTION 4.4

**VAE-USFlow** Before we outline the architecture, we briefly review the original VAE formulation: Consider a generative model with latent variables $\hat{p}_X(x) = \int \hat{p}_{X|Z}(x \mid z)p_Z(z)dz$ together with an amortized posterior approximation $\hat{p}_{Z|X}(z \mid x)$. The system is optimized by maximizing the expected evidence lower bound (ELBO) over the data distribution:

$$\mathrm{ELBO}(x; \hat{p}_{Z|X} \| \hat{p}_{Z|X}) = \ln \hat{p}_X(x) - \mathrm{KL}(\hat{p}_{Z|X}(z \mid x) \| \hat{p}_{Z|X}(z \mid x))$$

$$= \mathbb{E}_{z \sim \hat{p}_{Z|X}(z|x)} \left[ \ln \hat{p}_{X|Z}(x \mid z) - \ln \hat{p}_{Z|X}(z \mid x) + \ln p_Z(z) \right] \quad (7)$$

$$= \mathbb{E}_{z \sim \hat{p}_{Z|X}(z|x)} \left[ \ln \hat{p}_{X|Z}(x \mid z) \right] - \mathrm{KL}(\hat{p}_{Z|X}(z \mid x) \| p_Z(z)). \quad (8)$$

Typically, the latent prior $p_Z(z)$ is fixed as a simple distribution, e.g. $\mathcal{N}(0, I)$. From Equation 8 we see that when $p_Z(z)$ is learnable, maximizing the expected ELBO adapts the prior to the expected variational posterior, which is the true prior under the joint $\hat{p}_{X,Z}(x, z) = p_X(x)\hat{p}_{Z|X}(z|x)$ Tomczak & Welling (2018).

We are going to decompose our task of aligning the data in a compressed latent space into two tasks. The encoder (amortized variational posterior) encodes the instances in the compressed space, and a prior given by a trainable USF further aligns the densities with the norms in a secondary latent space. The ELBO objective avoids collapse in the primary space while the flow prior grants more flexibility to the feature encoder.

Our joint model $\hat{p}_{X,Z}^{\zeta,\eta}(x, z) = \hat{p}_{X|Z}^{\zeta}(x \mid z)p_Z^{\eta}(z)$ uses the following generative process:

$$B_{\mathrm{prior}} \sim \mathcal{N}(0, I),$$

$$Z \sim \phi_\eta^{-1}(B_{\mathrm{prior}}),$$

$$X \mid Z \sim \mathcal{N}(\mathrm{dec}_\zeta(z), \sigma_{\min}^2 I),$$

with $\sigma_{\min} > 0$ sufficiently small. The variational posterior approximation $\hat{p}_{Z|X}^{\eta}(z \mid x)$ is defined as:

$$Z \mid X \sim \mathcal{N}(\mathrm{enc}_\theta^\mu(x), \mathrm{enc}_\theta^\Sigma(x))$$

Using Equation 7, we optimize the model end-to-end via:

$$\max_{\zeta,\eta,\theta} \mathbb{E}_{x \sim \mathcal{D}} \mathbb{E}_{z \sim q_\theta(z|x)} \left[ \ln \mathcal{N}\left(x; \mathrm{dec}_\zeta(z), \sigma_{\min}^2 I\right) \right.$$

$$- \ln \mathcal{N}\left(z; \mathrm{enc}_\theta^\mu(x), \mathrm{enc}_\theta^\Sigma(x)\right)$$

$$\left. + \ln \mathcal{N}\left(\phi_\eta(z); 0, I\right) + \ln \left| \det \frac{\partial \phi_\eta}{\partial z}(z) \right| \right].$$

We provide an implementation of the proposed architecture, which we call VAE-USFlow. The model is based on wide-ResNet50 encoder (initialized with a pretrained weights) with a final fully connected layer to project to the desired latent dimension and a simple decoder, which is based on transposed convolutions. The number of layers of the decoder is based on the compression ratio of latent space. As flow prior, we employ the USFlows architecture and use the NonUSFlows architecture for baseline comparison (VAE-NonUSFlow, with softplus as affine activation and clamping at 2.0, analogous to Anomalib defaults). We train both variants on the MVTec datasets. Figure 11 in Appendix F.3 shows the mean image AUC-ROC performance and standard deviation over three runs for the individual classes.

## C  GAUSSIAN MIXTURE EXPERIMENT (SECTION 4.3)

The parameters of the asymmetric Gaussian mixture distribution for dimensionality $d = 2, 8, 32, 128$ are given as follows:

- $\mu_1 = \mathbf{1}_d, \mu_2 = -\mathbf{1}_d$
- $\Sigma_1 = I, \Sigma_2 = \mathrm{diag}(\theta_d),$

where $\mathbf{1}_d$ denotes the all ones vector in $d$ dimensions and $\theta_d \in \mathbb{R}^d$ is chosen such that half of the entries are $5.0$ and half are $1/2$. This asymmetric choice of covariance matrices is deliberate in order to incentivize inhomogeneous volume transfer via $\det \Sigma_1 = 1$ and $\det \Sigma_2 = 2.5^{\frac{d}{2}}$. For each model architecture and dimension, we conduct a hyperparameter optimization with 10 runs. For the flow architectures, we use (Non)USFlows with 2-10 coupling blocks and 2-3 layers per conditioner. For Deep SVDD, we use MLP encoders with depth 2–6 following a decreasing schedule based on the input dimension $d$ (i.e., $[d, d], [d, 2d, d], ... ,[d, 16d, 8d, 4d, 2d, d]$), forgoing bias terms and bounded activations as discussed in the preliminaries. We select the best-performing model per setting by validation loss (one-class for DeepSVDD, NLL for flows). Additionally, we optimize non architecture specific parameters such as learning rate and batch size. The entire experiment configuration is provided with the source code as YAML file.

Figure 5 plots the true likelihoods against the latent norms for DeepSVDD, USFlow and NonUSFlow. As one can see, only USFs consistently remain a monotonous relationship between data likelihoods and latent norms, which is witnessed by the Spearman $\rho$ and the Kendall $\tau$. Figure 6 plots the estimated likelihoods against the true data likelihoods. We see that USFlows and NonUSFlows are likewise able to model the data likelihoods relative accurately, despite differences in latent space alignment. Figure 2 visualizes the latent alignment for the 2D case. Figure 7 and Figure 8 visualize the volume transfer for the NonUSFlows model and the USFlows model, respectively. We encountered some numerical instabilities when working with NonUSFlows in 128 dimensions. See Section D for details.

## D  TRAINING INSTABILITIES OF NONUSFLOWS (SECTION 4 AND 5)

During our experiments, we consistently observed numerical instabilities when training flows that employ affine coupling and LU-decomposed affine transforms on high dimensional data, which lead to failed training runs. An investigation revealed that locally exploding determinants cause the problem. The issue becomes more pronounced with larger networks. While the problem is known when working with affine coupling, it seems to get amplified by the LU layers. We performed limited parameter tuning for critical parameters such as the affine clamping (which cuts the multiplicative part of the output of the affine coupling layers to a predefined range) but we couldn't eliminate the instabilities completely. We assume that this combination requires very careful, potentially problem specific, tuning of hyper parameters such as the prior scale, the affine clamping and parameter initialization, which we couldn't perform during our experiments. In the following, we summarize instabilities that we encountered during our experiments. Note that we observed no such instabilities when working with the USFlows architecture nor when working with affine coupling layers and affine transforms with determinant one (cf. Section 5), such as householder transforms or permutations (originally implemented in Anomalib). It is also noteworthy that the VAE-USFlow experiments and the ablation experiment do not share a common code basis since we implemented additive couplind and LU transforms in Anomalib independently.

**Gaussian Mixture Experiment**   During hyperparameter optimization, we encountered two failed runs for NonUSFlows architecture, both for models using 10 coupling blocks. Nevertheless, the the resulting model still proved to be very competitive w.r.t. the likelihood-likelihood comparison, which is why we accepted the results of the experiments. We did not observe any problems for the other dimensions.

**VAE-NonUSFlows**   VAE-NonUSFlows shows very instable behavior and clearly suboptimal outcomes on MVTec. Figure 11 in the main section documents failed runs and worse-than-random performance in detail.

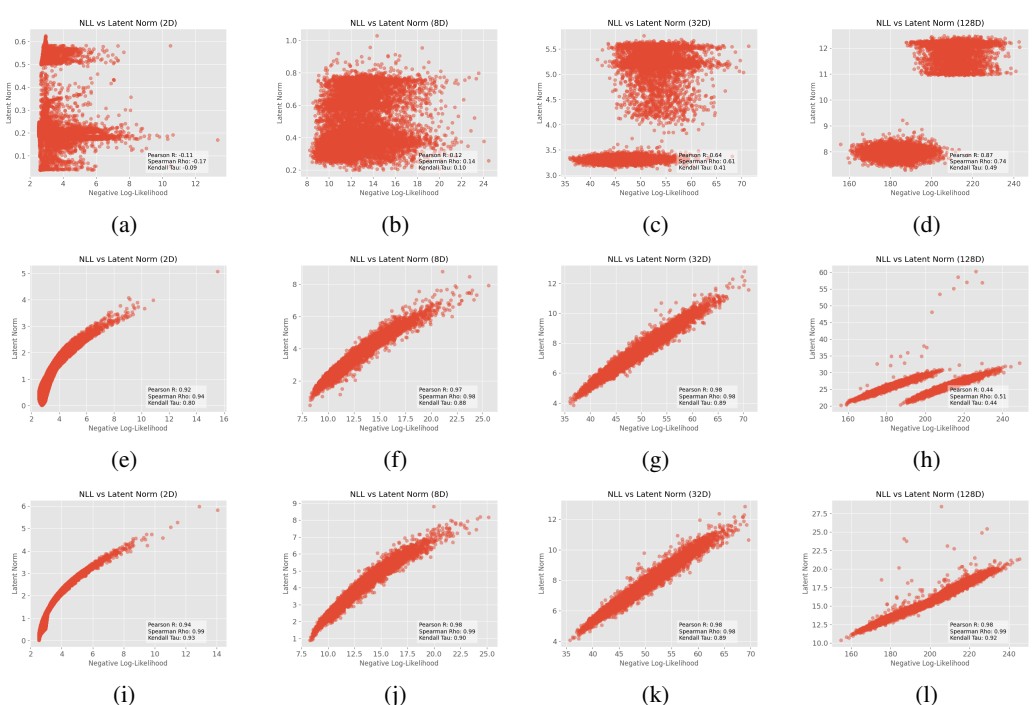

Figure 5: Scatter plots of the true log-likelihoods against the latent norms for the Gaussian mixture model experiments (2, 8, 32, 128 dimensions). (a) – (d): Deep SVDD (e) – (h): Non-uniformly scaling flow (i) – (l) Uniformly scaling flow. Uniformly scaling flows maintain a monotonic relationship between latent norm and data density, while for non-uniformly scaling flows remain some monotonic relationship but it can be broken in unpredictable ways due to the influence of the Jacobian determinant in the change of variables formula (see Figure 2 and Figure 7 for a 2D visualization). For Deep SVDD, the alignment initially appears random and splits into homogeneous clusters with increasing dimensionality.

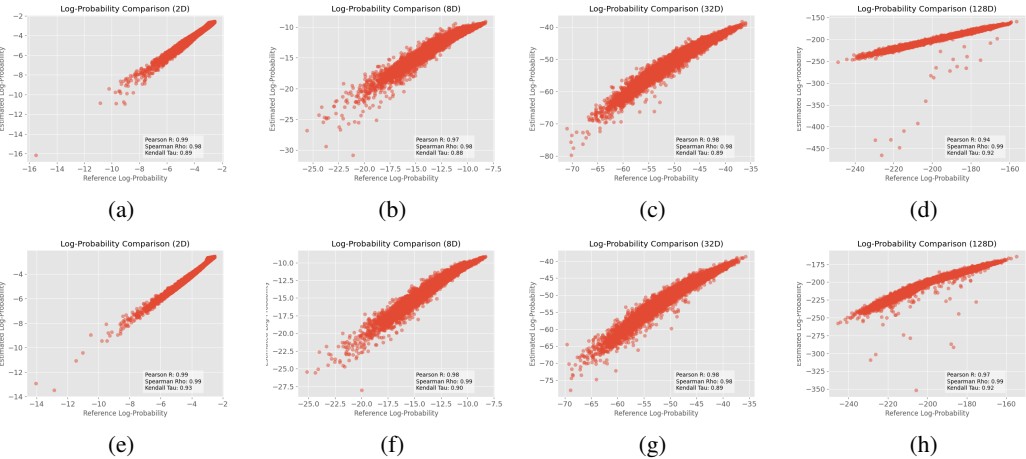

Figure 6: Scatter plots of the true log-likelihoods against the estimated log-likelihoods for the Gaussian mixture model experiments (2, 8, 32, 128 dimensions). (a) – (d): Non-uniformly scaling flow (e) – (h): Uniformly scaling flow. While we see clear differences in latent space alignment, especially for 128 dimensions, the quality of the estimated likelihoods remains roughly comparable among the flow models.

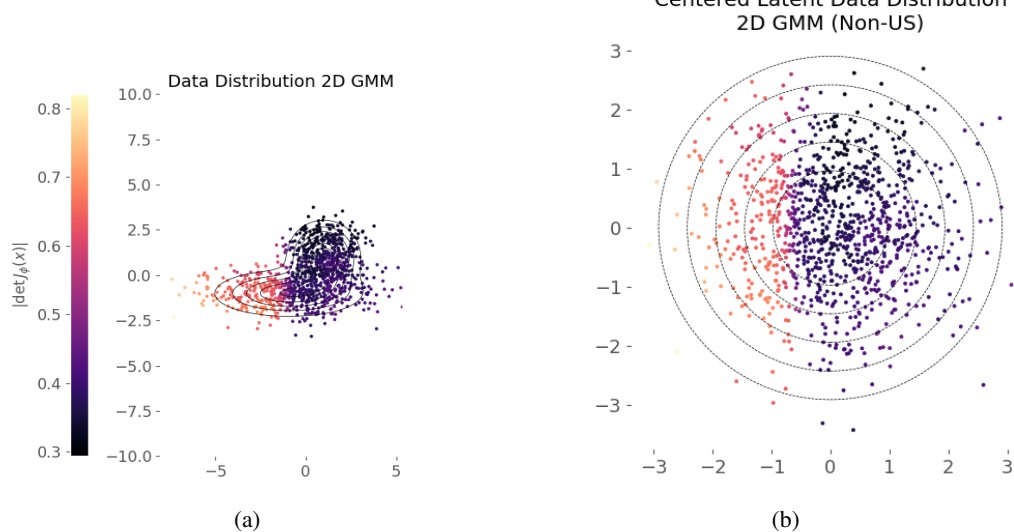

(a)                                                 (b)

Figure 7: Visualization of the transformation's volume change in different areas of the data distribution. **Left:** Sample and contours of the true data distribution. The determinant of the transformation Jacobian is color coded. **Right:** Sample and ideal contours of centered data latents of the **non-uniformly scaling flow**. The dashed contours show the contour lines of the flows base distribution (centered, $\sigma - 3\sigma$). The color of each sample encodes the absolute value of the determinant of the transformation Jacobian; The discrepancies in density alignment (see Figure 2c) are corrected by the variable volume change of the transformation so that the estimated likelihoods are still comparable to the ones we obtained from the uniformly scaling flow (see Figure 6a).

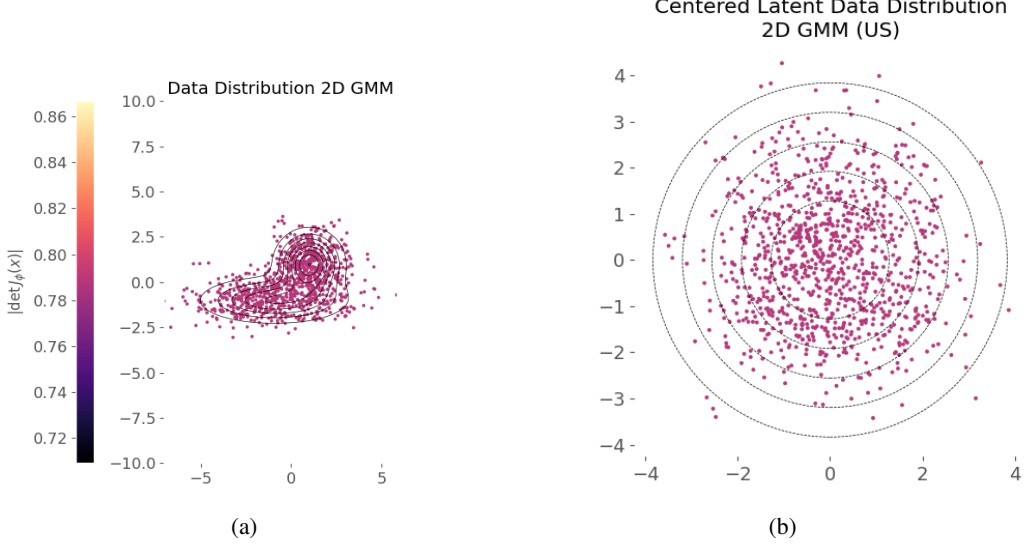

(a)                                                 (b)

Figure 8: Contour Plots for the 2D Gaussian mixture experiment. **Left:** Sample and contours of the true data distribution. The determinant of the transformation Jacobian is color coded. **Right:** Sample and ideal contours of centered data latents of the **uniformly scaling flow**. The dashed contours show the contour lines of the flows base distribution (centered, $\sigma - 3\sigma$). The color of each sample encodes the absolute value of the determinant of the transformation Jacobian. Since the determinant of the transformation Jacobian is constant by design, we see a uniform scaling, as expected.

**Ablation Study** We also encountered severe instabilities when using affine coupling and LU transforms in the ablation, in particular the different runs converged to the exact same (suboptimal) solution. Since we could not resolve the issue, we excluded the combination from the ablation. The obtained results on MVTec AD for the U-Flow architecture are reported in Figure 9.

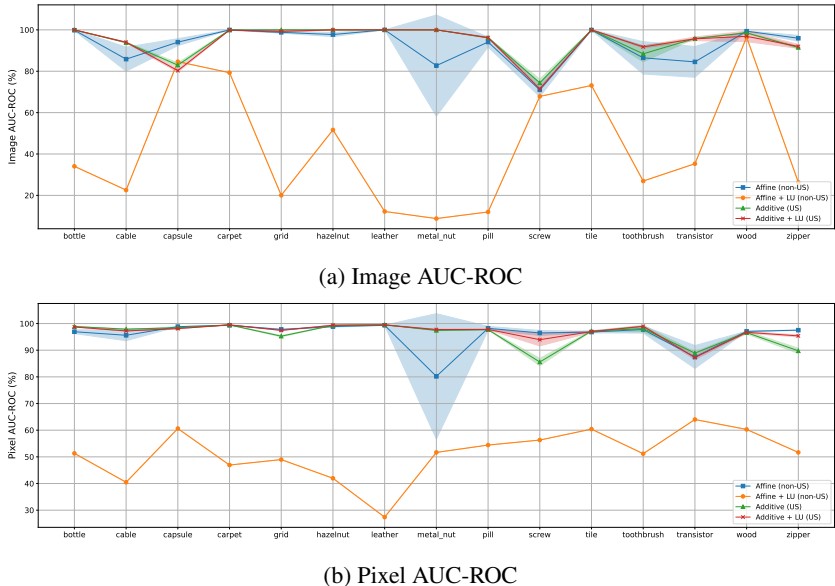

(a) Image AUC-ROC

(b) Pixel AUC-ROC

Figure 9: MVTec AD per-class mean (a) image- and (b) pixel-level AUC-ROC (%) ± stdev. across 3 runs for the U-Flow architecture when using different flow layers.

# E   FLOW ARCHITECTURE HYPERPARAMETERS (SECTION 5)

**CFlow** Feature maps are drawn from a Wide ResNet-50-2 backbones layers 2–4. The conditional normalizing flow consists of eight coupling blocks, each conditioned on a learned 128-dimensional positional embedding. To prevent numerical instability, the coupling network's log-scale outputs are clipped to $\pm 1.9$ before exponentiation. The learning rate is $1 \times 10^{-4}$ without weight decay.

**FastFlow** A ResNet-18 backbone feeds into eight alternating $1 \times 1$ and $3 \times 3$ convolutional coupling layers. The internal scale-and-shift subnetworks maintain the same channel width as their inputs, balancing expressivity with efficiency. Training uses a learning rate of $1 \times 10^{-3}$ and weight decay $1 \times 10^{-5}$.

**U-Flow** A MS-CaiT backbone produces a U-shaped, multi-scale feature pyramid. Four 2D affine coupling steps at each scale use full-width subnet channels to preserve detailed spatial information. Log-scale predictions are clamped to $\pm 2.0$ for stability. The initial learning rate is $1 \times 10^{-3}$ with weight decay $1 \times 10^{-5}$ and a linear decay to $40\%$ over 25,000 iterations.

# F   ADDITIONAL EXPERIMENTAL RESULTS (SECTION 5)

Here we report the full tables for the results discussed in Section 5.2.2 for image- and pixel-level AUC-ROC on MVTec AD and VisA the performance of the VAE-USFlow and Non-USFlow on MVTec AD in image-level AUC-ROC, as well as the pixel-level ablation counterpat to Section 5.2.3 .

## F.1   DETAILED RESULTS FOR MVTEC AD AND VISA

Tables 2, 3, 4 and 5 show detailed results for image- and pixel-level AUC-ROC on the MVTec AD and VisA datasets. The USF variants show improved average performance and stability.

| Class | BASE FLOW | | | USF | | |
|---|---|---|---|---|---|---|
| | FastFlow | U-Flow | CFlow | FastFlow | U-Flow | CFlow |
| bottle | **100.0±0.0** | 99.8±0.3 | 90.7±8.9 | 98.9±0.2 | **100.0±0.0** | **100.0±0.0** |
| cable | 91.8±0.8 | 85.8±7.1 | 85.7±6.8 | 71.0±2.3 | **94.0±0.3** | 86.2±1.1 |
| capsule | 88.9±1.7 | **94.1±2.1** | 83.5±7.9 | 84.7±1.2 | 80.3±0.9 | 87.2±0.6 |
| carpet | 98.0±0.6 | **100.0±0.0** | 78.4±8.4 | 96.8±0.2 | 99.9±0.0 | 95.9±1.3 |
| grid | 97.7±0.8 | 98.8±0.5 | 97.7±1.5 | 95.0±0.8 | **99.3±0.9** | 72.1±1.1 |
| hazelnut | 78.0±5.7 | 97.7±1.2 | 65.4±26.4 | 99.5±0.1 | **100.0±0.0** | **100.0±0.0** |
| leather | **100.0±0.0** | **100.0±0.0** | 96.4±5.1 | 99.7±0.1 | **100.0±0.0** | **100.0±0.0** |
| metal_nut | 97.1±1.4 | 82.7±29.9 | 32.6±42.6 | 95.5±0.2 | **100.0±0.0** | 99.3±0.2 |
| pill | 92.5±0.5 | 94.1±2.9 | 84.5±4.0 | 88.2±1.0 | **96.3±0.4** | 84.3±3.7 |
| screw | 67.7±2.4 | 71.0±4.1 | 83.8±10.7 | **83.9±1.2** | 71.5±0.8 | 61.9±4.7 |
| tile | 97.8±0.1 | 99.9±0.1 | 40.3±8.5 | 94.1±0.9 | **100.0±0.0** | 100.0±0.0 |
| toothbrush | 72.3±5.5 | 86.5±9.6 | 35.3±26.1 | 92.7±0.6 | 91.8±0.6 | **96.9±1.5** |
| transistor | 94.3±1.9 | 84.5±9.1 | 72.7±20.1 | 84.9±0.6 | **95.7±1.1** | 86.1±2.4 |
| wood | 98.1±0.3 | **99.4±0.1** | 53.6±7.4 | 98.2±0.2 | 96.9±3.2 | 99.1±0.1 |
| zipper | 95.9±0.1 | **95.9±1.7** | 86.2±7.3 | 81.9±3.2 | 92.0±1.1 | 92.8 ±0.2 |
| Average | 91.3±1.5 | 92.7±4.6 | 72.5±12.8 | 91.0±0.9 | 94.5±0.6 | 90.8±1.1 |
| #Best | 2 | 5 | 0 | 1 | 9 | 4 |

Table 2: Mean image AUC-ROC and stdev. values (%) per MVTec AD class, averaged over 3 runs per configuration. Highest mean value per row in bold, second-highest underlined. "#Best" shows how many times each column had the highest mean value.

| Class | BASE FLOW | | | USF | | |
|---|---|---|---|---|---|---|
| | FastFlow | U-Flow | CFlow | FastFlow | U-Flow | CFlow |
| bottle | 97.4±0.2 | 96.9±0.9 | 97.7±0.2 | 97.8±0.1 | **98.7±0.1** | 98.3±0.0 |
| cable | 94.9±0.2 | 95.6±2.5 | 94.9±1.4 | 93.8±1.1 | **97.2±0.1** | 95.2±0.1 |
| capsule | 97.7±0.6 | **98.8±0.0** | 98.6±0.2 | **98.8±0.0** | 98.1±0.0 | **98.8±0.0** |
| carpet | 98.6±0.1 | 99.3±0.1 | 98.8±0.1 | 98.7±0.0 | **99.5±0.0** | 98.8±0.0 |
| grid | 98.0±0.4 | 97.8±0.1 | 97.4±0.1 | **98.3±0.1** | 97.4±0.1 | 96.7±0.1 |
| hazelnut | 93.3±1.0 | 98.9±0.2 | 97.0±0.7 | 98.3±0.0 | **99.4±0.0** | 98.5±0.0 |
| leather | **99.6±0.0** | 99.3±0.1 | 99.4±0.1 | 98.9±0.2 | 99.5±0.1 | 99.1±0.0 |
| metal_nut | 96.0±0.3 | 80.2±28.8 | 92.3±5.1 | 97.3±0.1 | **97.8±0.2** | 97.5±0.1 |
| pill | 96.9±0.4 | **98.2±0.9** | 97.2±0.4 | 97.7±0.0 | 97.8±0.1 | 98.0±0.0 |
| screw | 84.5±1.9 | 96.5±1.1 | 97.5±0.4 | **98.3±0.1** | 93.9±2.9 | 96.8±0.1 |
| tile | 94.4±0.3 | 96.8±0.4 | 94.2±0.7 | 92.1±0.2 | **97.0±0.3** | 95.8±0.0 |
| toothbrush | 96.3±0.3 | 97.8±1.6 | 97.6±0.3 | 98.9±0.0 | **99.0±0.1** | 98.4±0.0 |
| transistor | **96.4±0.4** | 87.5±5.2 | 84.6±2.2 | 90.8±0.1 | 87.3±0.9 | 85.9±0.2 |
| wood | 95.3±0.3 | **97.1±0.1** | 87.4±2.3 | 94.2±0.2 | 96.7±0.4 | 94.3±0.0 |
| zipper | **97.7±0.2** | 97.5±0.2 | 96.7±1.4 | **97.7±0.2** | 95.4±0.4 | 97.0±0.0 |
| Average | 95.8±0.4 | 95.9±2.8 | 95.4±1.0 | 96.8±0.2 | 97.0±0.4 | 96.6±0.0 |
| #Best | 3 | 3 | 0 | 4 | 7 | 1 |

Table 3: Mean pixel AUC-ROC and stdev. values (%) per MVTec AD class, averaged each over 3 runs. Highest value per row in bold, second-highest underlined. "#Best" shows how many times each column had the highest value.

| Class | BASE FLOW | | | USF | | |
|---|---|---|---|---|---|---|
| | FastFlow | U-Flow | CFlow | FastFlow | U-Flow | CFlow |
| candle | **96.5±0.3** | 92.7±1.5 | 91.2±2.8 | 84.9±1.1 | 84.1±2.0 | 94.6±0.4 |
| capsules | 73.9±1.9 | 74.2±6.5 | 77.6±14.8 | 75.5±0.4 | **79.7±0.8** | 70.8±5.4 |
| cashew | 88.4±2.2 | 93.7±1.5 | **96.7±1.2** | 90.6±0.7 | 94.7±0.5 | 95.9±0.6 |
| chewinggum | 98.2±0.3 | 98.5±0.5 | **99.6±0.1** | 99.4±0.1 | 98.6±0.3 | **99.6±0.1** |
| fryum | 89.5±5.0 | 88.5±1.5 | 52.2±18.6 | **92.8±0.5** | 86.0±0.2 | 84.9±0.5 |
| macaroni1 | 87.8±2.9 | 79.8±22.2 | 71.4±10.7 | 88.8±0.5 | 77.6±0.3 | **89.3±0.2** |
| macaroni2 | 67.8±2.5 | 63.5±13.6 | 78.6±18.2 | **80.5±0.5** | 74.8±1.2 | 71.6±1.4 |
| pcb1 | 87.9±1.4 | 76.8±15.2 | 87.8±5.4 | 83.8±1.4 | 83.1±0.2 | **92.7±0.4** |
| pcb2 | **87.7±0.8** | 80.8±7.1 | 82.9±4.7 | 85.7±0.9 | 76.6±0.4 | 86.3±0.8 |
| pcb3 | **86.9±1.3** | 80.5±8.8 | 71.6±11.5 | 81.4±0.3 | 85.5±1.2 | 79.5±0.6 |
| pcb4 | 96.2±0.2 | 90.4±1.8 | 96.1±1.2 | 92.5±0.6 | 88.7±0.5 | **97.3±0.4** |
| pipe_fryum | 94.9±1.7 | 95.3±0.6 | 84.8±9.9 | 96.7±0.1 | 97.9±0.1 | **98.6±0.1** |
| Average | 88.0±1.7 | 84.6±6.7 | 82.5±8.3 | 87.7±0.6 | 85.6±0.6 | 88.4±0.9 |
| #Best | 3 | 0 | 2 | 2 | 1 | 5 |

Table 4: Mean image AUC-ROC and stdev. values (%) per VisA class, averaged each over 3 runs. Highest value per row in bold, second-highest underlined. "#Best" shows how many times each column had the highest value.

| Class | BASE FLOW | | | USF | | |
|---|---|---|---|---|---|---|
| | FastFlow | U-Flow | CFlow | FastFlow | U-Flow | CFlow |
| candle | 98.0±0.4 | 99.0±0.1 | 99.0±0.0 | **99.1±0.0** | 94.8±3.5 | 99.0±0.0 |
| capsules | 98.0±0.1 | 95.7±3.5 | 96.5±0.7 | **99.2±0.0** | 98.2±0.1 | 97.2±0.0 |
| cashew | 98.2±0.3 | **99.4±0.1** | 98.3±0.3 | 98.8±0.0 | 99.3±0.1 | 98.4±0.0 |
| chewinggum | 98.9±0.1 | 99.2±0.1 | 99.0±0.2 | **99.4±0.0** | 99.3±0.2 | **99.4±0.0** |
| fryum | 84.6±4.6 | 95.5±0.5 | **97.1±0.8** | 97.1±0.0 | 96.3±0.1 | 97.0±0.1 |
| macaroni1 | 97.8±0.7 | 94.0±9.1 | 97.3±0.6 | **99.7±0.0** | 88.3±8.4 | 99.1±0.0 |
| macaroni2 | 94.5±0.3 | 81.9±20.2 | 97.1±1.3 | **99.1±0.0** | 82.6±8.0 | 98.3±0.1 |
| pcb1 | 99.1±0.1 | 95.1±7.1 | 99.3±0.1 | 99.2±0.0 | **99.4±0.0** | 99.3±0.0 |
| pcb2 | 96.7±0.4 | 97.2±0.7 | 97.4±0.4 | **98.3±0.1** | 96.3±0.3 | 96.7±0.0 |
| pcb3 | 96.6±0.2 | 97.2±0.9 | 97.0±0.4 | **98.7±0.0** | 96.3±1.4 | 97.3±0.0 |
| pcb4 | 97.5±0.3 | 96.2±2.5 | 97.2±0.6 | 97.2±0.1 | **98.5±0.0** | 97.7±0.1 |
| pipe_fryum | 96.0±1.0 | 98.9±0.6 | 98.8±0.1 | **99.2±0.0** | **99.2±0.0** | 99.0±0.0 |
| Average | 96.3±0.7 | 95.8±3.8 | 97.8±0.5 | 98.8±0.0 | 95.7±1.8 | 98.2±0.0 |
| #Best | 0 | 1 | 1 | 9 | 3 | 1 |

Table 5: Mean pixel AUC-ROC and stdev. values (%) per VisA class, averaged each over 3 runs. Highest value per row in bold, second-highest underlined. "#Best" shows how many times each column had the highest value.

## F.2 ABLATION STUDY (MVTEC AD, PIXEL)

Figure 10 reports the pixel-level AUC-ROC ablation, showing the same tendencies as in Section 5.2.3.

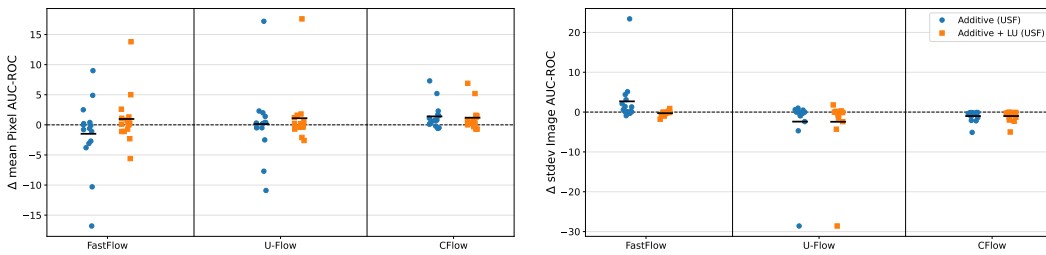

(a) $\Delta$ mean pixel AUC-ROC (higher is better).   (b) $\Delta$ stdev pixel AUC-ROC (lower is better).

Figure 10: Ablation relative to the affine baseline per MVTec AD class. The dashed $y{=}0$ lines mark the affine baseline; the black horizontal mark is the class-wise mean $\Delta$. *Additive (USF)* swaps affine for additive coupling; *Additive + LU (USF)* additionally inserts LU-parameterized affine transforms.

## F.3 VAE-FLOW ON MVTEC AD

An evaluation plot of the VAE-Flow performance is given in Figure 11.

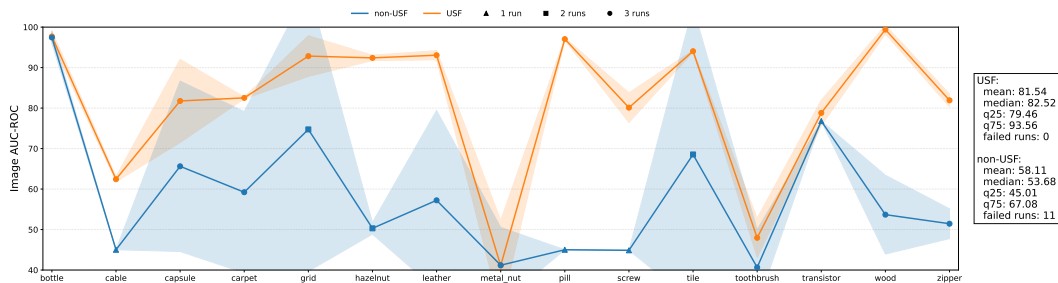

Figure 11: MVTec AD per-class mean image AUC-ROC (%) $\pm$ stdev. for the VAE-Hybrid setting. USF consistently achieves higher average performance and markedly improved run-to-run performance stability. Only for non-USF we frequently encountered training failures due to numerical instabilities (see Appendix D). We indicate the number of valid runs (max 3) per class with distinct markers.

