# OpenReview forum: "On Uniformly Scaling Flows: A Density-Aligned Approach to Deep One-Class Classification"
_ICLR.cc/2026/Conference — Submitted to ICLR 2026_

### Official Review · Reviewer_U6Ni · 2025-10-17

**Soundness:** 2
**Presentation:** 3
**Contribution:** 2
**Rating:** 4
**Confidence:** 4

**Summary:**

The paper connects Deep SVDD and normalizing flows for anomaly detection by introducing a subclass called Uniformly Scaling Flows (USFs), where the Jacobian determinant is constant across inputs. Under this setup, the maximum-likelihood objective of a flow becomes equivalent to the Deep SVDD loss, offering a simple link between one-class classification and flow-based modeling. The authors argue that this keeps the stability and invertibility benefits of flows while avoiding the collapse issues of Deep SVDD. In experiments, they replace affine coupling layers with additive USF layers in existing flow-based detectors (FastFlow, CFlow, U-Flow) using frozen pretrained backbones. Results on MVTec AD and VisA show that the USF variants reach similar or slightly better accuracy, with improved training stability and less sensitivity to initialization. The paper positions USFs as a lightweight and theoretically grounded alternative to standard flow-based anomaly detectors.

**Strengths:**

- The paper formalizes a clear theoretical link between Deep SVDD and a subclass of normalizing flows with constant Jacobian determinants, unifying two separate paradigms of anomaly detection through an analytical derivation.
- The paper reinforces previous observations that the log-determinant term can dominate likelihood in flow-based detectors, and systematically validates that removing input-dependent volume terms leads to more stable training.
- The paper includes a range of controlled experiments and ablations that clearly isolate the effect of the proposed change, making the empirical findings easy to interpret.
- The paper is generally easy to follow, with clean derivations and consistent notation. The theory-to-experiments flow is coherent, and the ablation results are presented in a readable and well-organized manner.

**Weaknesses:**

- The paper builds its main theoretical motivation on Deep SVDD, a 2018 one-class method that has largely fallen out of use in both image and tabular anomaly detection. Modern benchmarks (e.g., ADBench, MVTec, VisA) consistently show that simple feature-based or transformer-based approaches, contrastive learning, and diffusion-based detectors far outperform Deep SVDD, which is now mostly cited for historical context. As a result, grounding the paper’s core theory around an outdated baseline places it in an awkward position. The conceptual bridge it draws may be elegant, but its practical impact is limited unless validated against current methods.
-  The paper frames Uniformly Scaling Flows (USFs) as a key innovation for removing input-dependent volume effects, but constant-Jacobian / volume-preserving flows have long existed in the literature (e.g., NICE; FlowSVDD; OneFlow), and have already been applied specifically to anomaly detection to avoid SVDD-style collapse. As such, the only substantial novelty lies in the formal SVDD–USF MLE equivalence and the controlled additive-vs-affine ablations, not in the idea of using constant-determinant flows itself.
- The empirical section mostly replaces affine coupling with additive coupling in pre-existing architectures (FastFlow, CFlow, U-Flow) using pretrained frozen features. The practical contribution is thus modest. It is more of an architectural ablation than a fundamentally new detection framework. The theoretical bridge to Deep SVDD does not clearly lead to a new practical algorithm beyond this substitution.
- Each baseline uses a different pretrained backbone (ResNet-18, WRN-50-2, CaiT), so absolute numbers are not directly comparable across architectures; only within-architecture swaps are fair. A shared backbone control would strengthen the empirical claims.
- In the main text, in Proposition 1, $F_\alpha(x)=x/(\alpha||x||)$ gives $||F_\alpha(x)||\equiv 1/\alpha$ for all $x\neq0$, so the claimed monotonicity $||x||\uparrow\Rightarrow||F_\alpha(x)||\downarrow$ is false; yet the loss is then (incorrectly) written as $\alpha^{-2}\mathbb{E}||x||^{-2}$. In Appendix B.4, $F_\alpha$ is redefined as the scalar $1/(\alpha|x|)$ but treated as vector-valued, creating a type mismatch with the Deep-SVDD loss $\mathbb{E}||F(X)-c||^2$. A minimal fix is the vector-valued radial inversion $F_\alpha(x)=x/(\alpha||x||^2)$ with $F_\alpha(0)=0$, which yields the intended ordering and $L=\alpha^{-2}\mathbb{E}\big[1/||x||^2\big]=1/\big(\alpha^2(d-2)\big)$ for $d>2$.
- The citations should be in parentheses using \citep.

**Questions:**

- How sensitive are the reported gains to the choice of pretrained backbone? Would the same improvements hold if all methods shared an identical architecture or were trained from scratch?
- How does the proposed USF formulation differ from prior work such as FlowSVDD, which also uses constant-Jacobian (volume-preserving) flows for anomaly detection and explicitly argues that this avoids hypersphere collapse?
- How does the improved flow baselines compared to state-of-the-art non-flow baselines?
- The experiments rely on MVTec AD and VisA, both image datasets, even though all features come from pretrained ImageNet backbones. Why restrict the evaluation to images? If the method is architecture-agnostic, testing on tabular anomaly detection (e.g. ADBench) could better show its generality.

---

> ### Author Response · Authors · 2025-11-21
> **Response to Reviewer U6Ni**
>
> We thank the reviewer for the thorough review and positive comments on presentation. We address your concerns below.
>
> Addressing Weaknesses
> W1: "Deep SVDD is an outdated baseline..."
> We respectfully disagree. Deep SVDD variants remain relevant in current research (e.g., Zhang et al. ICLR 2024 builds on Deep SVDD principles). Our analysis shows that addressing its latent space alignment issues improves modern AD methods. The Deep SVDD connection explains why USFs work, not that Deep SVDD itself is state-of-the-art.
>
> Our experiments demonstrate practical impact: USF substitution improves CFlow (72.5→90.8%), U-Flow (92.7→94.5%), and reduces variance across methods on modern benchmarks against modern flow-based detectors.
>
> W2: "Constant-Jacobian flows have long existed..."
> Our novelty extends beyond the equivalence:
>
> Different training: FlowSVDD/OneFlow use custom losses; we show standard MLE suffices.
>
> Stronger guarantees: Collapse prevention via KL-divergence minimization.
>
> Alignment analysis: We prove monotonic density-norm alignment.
>
> Pathology identification: We characterize density inversion in Deep SVDD.
>
> W3: "Different backbones mean numbers not comparable."
> This is intentional. We used default backbones to reflect out-of-the-box deployment. We claim within-architecture improvements are robust, not that cross-architecture numbers are comparable.
>
> W4: Mathematical error in Proposition 1
> We apologize for this error. The intended definition is $F_\alpha(x) = 1/(\alpha|x|)$ with $F_\alpha(0)=0$. The correction $x/(\alpha|x|^2)$ is equivalent in loss and ranking. We will correct this.
>
> W5: Citations should use \citep
> We will revise the citation formatting.
>
> Addressing Questions
> Q1: Sensitivity to pretrained backbone?
> Backbones were determined by Anomalib defaults—no deliberate selection. We expect robustness to backbone choice and are validating this.
>
> Q2: How does USF differ from FlowSVDD?
>
> FlowSVDD uses specialized losses breaking density connection
>
> Our collapse prevention combines bijectivity with KL-divergence
>
> We analyze density-norm monotonicity not considered in FlowSVDD
>
> Q3: Comparison to non-flow SOTA?
> Our ablation isolates the USF effect within flow architectures. Non-flow comparisons would confound architectural differences. We will add contextual leaderboard comparisons.
>
> Q4: Why restrict to images?
> This reflects Anomalib's current scope. We are extending to tabular data to strengthen generality claims.

---

> > ### Comment · Reviewer_U6Ni · 2025-11-27
> >
> > I thank the authors for their response. I have reviewed your feedback and the current state of the manuscript.
> >
> > **Relevance of Deep SVDD**. I looked further into the literature, and your point about Deep SVDD’s continued relevance holds up. Recent work such as Zhang et al. (ICLR 2024), along with several other papers I found, confirms that hypersphere collapse is still an active research problem.
> >
> > **Theoretical Contribution vs. Empirical Validation**. I appreciate the elegance of the proposed theoretical unification. Connecting Deep SVDD to Normalizing Flows via the standard MLE objective (rather than the custom losses used in FlowSVDD or OneFlow) is a valuable contribution. However, to fully validate the practical utility of this "cleaner" theoretical formulation, it is essential to compare it empirically against those existing volume-preserving baselines. Since FlowSVDD and OneFlow are the direct competitors using constant-Jacobian flows, demonstrating that your MLE-based USF matches (or exceeds) their performance is necessary to show that this theoretical bridge translates into effective practice.
> >
> > **Experimental Scope and Saturation**. MVTec AD and VisA are restricted to industrial visual inspection and are arguably saturated benchmarks, making small performance gains less significant. Furthermore, performance on these datasets is heavily dependent on the frozen pretrained backbones used. To support a general claim about flow-based anomaly detection, the method should be evaluated on semantic anomaly detection or tabular data where the flow must learn the distribution from scratch. I note your promise to add tabular experiments, but they are currently missing.
> >
> > As there have been no changes to the manuscript text as of now, I cannot fully assess the quality of the promised corrections or the impact of the new experiments. I am maintaining my score for now, but I am willing to increase it if the promised changes are explicitly incorporated into the manuscript. To be clear, my main reservation is that the empirical claims are weak without these broader validations.

---

> > > ### Author Response · Authors · 2025-11-27
> > >
> > > We thank the reviewer for the positive reassessment and the constructive feedback. We acknowledge that the empirical validations are crucial.
> > >
> > > Currently, we are conducting additional experiments to address your concerns:
> > > 1.  **ADBench tabular tests** against FlowSVDD and OneFlow to demonstrate generality and direct competitive performance.
> > > 2.  **Re-runs of the industrial benchmarks** for all methods using a common backbone to ensure fair comparison.
> > >
> > > Due to the high computational demand of these experiments, the updated manuscript with all results and corrections will likely be ready towards the end of the rebuttal period. We are confident these additions will fully address your reservations and hope the final version will merit an increased score.

---

### Official Review · Reviewer_b6fW · 2025-11-01

**Soundness:** 2
**Presentation:** 2
**Contribution:** 2
**Rating:** 2
**Confidence:** 4

**Summary:**

This paper proposes Uniformly Scaling Flows (USFs), a restricted class of normalizing flows where the Jacobian determinant is constant across the input space. The authors show that maximum-likelihood training of such a flow can be reformulated as a Deep SVDD-style objective with an implicit regularization term preventing collapse. Experiments on replacing standard coupling layers in FastFlow, CFlow, and U-Flow with USF variants showed moderate AUROC improvements and notably reduced run-to-run variance on MVTec AD and VisA benchmarks.

**Strengths:**

1. The paper provides a clean mathematical connection between one-class classification and flow-based objectives.
2. This work empirically shows improved training stability across several flow architectures.
3. The idea of exploring Jacobian regularization may inspire follow-up work on stable density modeling.

**Weaknesses:**

1. The theoretical equivalence is mathematically valid but largely follows from simplifying the log-det term in standard flows. It does not lead to a new learning principle.
2. The main idea is an extension of the prior hybrid flow–SVDD and One-Flow works. The difference from these prior works is a mild variation rather than a substantive theoretical or algorithmic innovation.
3. Only modest empirical improvements are shown on many classes in the experiments.
4. The evaluation is limited to two industrial anomaly detection datasets, and the experimental comparisons are limited to flow-based methods. They didn't include the latest SOTA methods into the experimental comparisons.

**Questions:**

1. The derivations are internally consistent and reproduce known properties of flow-based AD models. However, the equivalence theorem is straightforward once the constant determinant assumption is applied, and the empirical validation does not seem to justify the claimed unification.
2. Section 5 on experiments reads like an afterthought and does not deeply analyze why stability is improved.

---

> ### Author Response · Authors · 2025-11-21
> **Response to Reviewer b6fW**
>
> We thank the reviewer for their feedback. While we respectfully disagree with the assessment, we appreciate the opportunity to clarify our contributions.
>
> Addressing Weaknesses
> W1: "The theoretical equivalence... does not lead to a new learning principle."
> We disagree. Our analysis reveals that USFs unify density estimation and latent space alignment under a single MLE objective. This is a novel insight with key implications:
>
> MLE training of USFs inherently implements a regularized Deep SVDD objective, provably preventing collapse.
>
> USFs guarantee monotonic alignment between data density and latent norm, a property absent in standard Deep SVDD or non-USF flows.
>
> We identify pathological solutions in Deep SVDD that invert densities, a deficiency unaddressed in prior work.
>
> This leads to a new design principle: prefer USF architectures for density-based anomaly detection as they naturally align likelihood and distance.
>
> W2: "The main idea is an extension of prior hybrid flow-SVDD..."
> This mischaracterizes our contribution. Prior works (FlowSVDD, OneFlow) use custom loss functions that break the connection to density estimation. Our key insight is that standard MLE training of USFs intrinsically provides the desired properties without custom objectives—a fundamentally different finding.
>
> W3: "Only modest empirical improvements are shown..."
> We demonstrate substantial improvements in stability (reduced variance) across all architectures and datasets, with notable accuracy gains (e.g., CFlow: +18.3 on MVTec; U-Flow: +1.8 on MVTec). Even where average gains appear modest, we see large improvements on difficult classes (e.g., FastFlow on 'screw': +16.2 points). These gains come from a simple drop-in replacement, showcasing the broad applicability of our insights in a saturated benchmark space.
>
> W4: "The evaluation is limited..."
> We acknowledge this and are extending experiments. Our ablation was designed to isolate the effect of the USF property itself; including non-flow methods would conflate architectural differences with our core contribution.
>
> Addressing Questions
> "The equivalence theorem is straightforward... empirical validation does not justify the claimed unification."
> While the algebra is simple, the implications are not. Our analysis shows the connection is profound: collapse avoidance stems from minimizing reverse KL-divergence, and the constant Jacobian leads to full monotonic alignment. Our empirical results confirm these theoretical insights translate to consistent stability and accuracy improvements.
>
> "Section 5... does not deeply analyze why stability is improved."
> The stability gains are directly motivated by our theory. Variable log-det terms can have high gradients in low-density regions, causing instability. The USF objective provides a more stable signal for tail estimation. We will add an explicit discussion connecting these points in the revision.

---

### Official Review · Reviewer_cPS3 · 2025-11-01

**Soundness:** 4
**Presentation:** 3
**Contribution:** 3
**Rating:** 6
**Confidence:** 4

**Summary:**

The theory is clear and careful: training a uniformly scaling flow (USF) with MLE reduces to a Deep-SVDD-style objective with an implicit weight regularizer, which explains why collapse is avoided. The constant-Jacobian view cleanly links density level sets to latent norms. Empirically, the drop-in USF swap improves stability strongly and often accuracy across MVTec AD and VisA on CFlow and U-Flow (e.g., ~72.5→90.8 and ~92.7→94.5), while FastFlow is roughly neutral. The scope is mainly image AD; dimensionality-reduction needs the proposed VAE hybrid. Overall, technically solid.

**Strengths:**

1. Clear theoretical link between USFs and Deep SVDD that explains behavior and avoids collapse.
2. Practical “drop-in” recipe that substantially reduces run-to-run variance; accuracy gains on CFlow/U-Flow are convincing.
3. Broad evaluation across datasets, metrics, and architectures; ablations isolate the US modification.
4. Writing is professional; related work coverage is adequate.

**Weaknesses:**

1. Architectural novelty is incremental (additive/volume-preserving flows are known); the novelty depends mostly on the connection and analysis.
2. Expressivity trade-off vs. affine coupling is under-analyzed; FastFlow gains are limited.
3. Evaluation scope is visual AD; no results on tabular/time-series or non-vision AD.
4, VAE-USF section is promising but under-evaluated (few settings, limited ablations).
5. Baselines emphasize flow variants; adding non-flow one-class/reconstruction baselines would strengthen the broader claim.

**Questions:**

1. Beyond isotropic Gaussian bases, do heavy-tailed or mixture bases preserve the same alignment benefits in USFs?
2. For FastFlow, why are gains small? Capacity, optimization, or interaction with 2D conv flows?
3. Did you actually apply the log-normal prior on det(J) during training, and how sensitive are results to its hyperparameters?
4. In VAE-USF, how did you set the reconstruction vs. likelihood weighting, and did varying latent dimension change the story?
5. Any path to dimension reduction without the VAE (e.g., relaxed invertibility or partial flows)?

---

> ### Author Response · Authors · 2025-11-21
> **Response to Reviewer cPS3**
>
> We thank the reviewer for the positive assessment (Rating 6, Soundness 4) and for recognizing our theoretical contributions and empirical validation. We address your specific questions and concerns below.
>
> Questions
>
> Q1: Beyond isotropic Gaussian bases, do heavy-tailed or mixture bases preserve the same alignment benefits in USFs?
>
> All uniformly scaling flows preserve density alignment in the sense that $p_X(x) < p_X(y) \Leftrightarrow p_{\phi(X)}(\phi(x)) < p_{\phi(X)}(\phi(y))$. However, to preserve alignment of the density with the latent norm (i.e., $p_X(x) < p_X(y) \Leftrightarrow \|\phi(x)\| > \|\phi(y)\|$), the base distribution needs to be radially monotonic—density must be a monotonic function of the distance from the center.
>
> Heavy-tailed radial distributions can preserve this property, while mixture bases would not guarantee monotonic alignment. However, we believe that such flexible base distributions might help produce solutions with higher fidelity in complex, multi-modal scenarios, which is an interesting direction for future work. We will clarify this in the revised manuscript.
>
> Q2: For FastFlow, why are gains small? Capacity, optimization, or interaction with 2D conv flows?
>
> The "small gains" statement is relative to the MVTec dataset and image-level metrics. On VisA, we observed much more pronounced gains for FastFlow, especially for pixel-level metrics (96.3→98.8%, Table 2). The architecture-specific behavior likely reflects interactions between the 2D convolutional structure and the flow modification, but a complete analysis across all dataset-architecture combinations is beyond the scope of this work given the combinatorial explosion (3 architectures × 2 datasets × 2 metric types).
>
> The key message from our ablation is that a simple, uniform architectural change—which aligns the objective with both density estimation and latent norm reasoning principles—consistently produces improved stability and often improved accuracy across a wide range of architectures, even in the highly optimized domain of industrial anomaly detection. We believe this demonstrates the high practical relevance of our theoretical analysis.
>
> Q3: Did you actually apply the log-normal prior on det(J) during training, and how sensitive are results to its hyperparameters?
>
> Yes, we applied a log-normal prior with default scale σ = 1.0 (library default) for all USF experiments. The training is typically not very sensitive to the scale hyperparameter in the range [0.5, 3.0], as the penalty converges toward zero during late-stage training.
>
> Exploding determinants are not common with our adjoint-action architecture (where affine transforms are applied as $A^{-1} \circ C \circ A$). However, if affine transforms were simply prepended before each coupling layer, exploding determinants would be frequent due to the multiplicative nature of determinants. We conducted limited ablations on the prior's influence for Section 5 architectures, showing slight improvements over the non-regularized variant. We will include these results in the revised appendix.
>
> Q4: In VAE-USF, how did you set the reconstruction vs. likelihood weighting, and did varying latent dimension change the story?
>
> We used β = 1.0 (standard VAE weighting) for the β-VAE component, following the default setting. We did not extensively explore different β values or varying latent dimensions in the current work, as our focus was on demonstrating proof-of-concept for dimensionality reduction with USFs.
>
> Q5: Any path to dimension reduction without the VAE (e.g., relaxed invertibility or partial flows)?
>
> This is an excellent question. There is growing work on injective networks that map from lower-dimensional to higher-dimensional spaces, which could offer an alternative path. While we believe this line of research may yield interesting candidates for anomaly detection, exploring it was beyond the scope of our current work. We will add a discussion of this promising direction in the revised manuscript.
>
> Addressing Weaknesses
>
> On limited evaluation scope (visual AD only): As mentioned in the general response, we are extending our experiments to additional domains. We will update the manuscript with results on non-image data in the coming days.
>
> On VAE-USF being under-evaluated: We agree and are conducting additional experiments.

---

### Author Response · Authors · 2025-11-21
**General Response to All Reviewers**

We sincerely thank all reviewers for their thoughtful and constructive feedback. We appreciate the recognition of our theoretical contributions and empirical validation. We address the common concerns raised across reviews below, with detailed responses to individual reviewers following.

### On Practical Value and Scope of Experiments

Several reviewers noted the limited experimental scope (image-only datasets) and requested broader validation. We acknowledge this limitation and are actively extending our experiments to address these concerns. Our survey of current flow-based anomaly detection libraries reveals that USF architectures are largely absent from mainstream implementations:

- **GANF** (Graph AD with NFs): Only non-USFs
- **STG-NF**: Contains both USF and non-USF variants
- **Anomalib**: Only non-USFs
- **SanFlow**: Only non-USFs

This gap underscores the practical value of our investigation. Since the fundamental differences in latent space alignment and their consequences for anomaly detection are currently under-investigated, our experiments clearly indicate that USF architectures have the potential to improve a large number of highly performant AD methods in a drop-in fashion. A well-grounded theory that connects density estimation and latent norm-based reasoning allows practitioners to make informed decisions during AD algorithm design, which can have substantial influence as our experiments demonstrate.

We note that recent theoretical work has analyzed conditions under which volume-preserving flows may be suboptimal, particularly for multi-modal distributions (Draxler et al., ICML 2024, "On the Universality of Volume-Preserving and Coupling-Based Normalizing Flows"). Interestingly, their analysis reveals that when the latent distribution is rotationally symmetric with density decreasing from the origin—exactly our setup with isotropic Gaussian bases—volume-preserving flows learn the same transport map as more expressive alternatives. This theoretical insight supports our empirical finding that USF substitution improves performance in the anomaly detection setting, where the base distribution satisfies these conditions.

### On Novelty Beyond Prior Work (FlowSVDD, OneFlow)

Multiple reviewers questioned the novelty relative to FlowSVDD and OneFlow. We emphasize that while these works employ volume-preserving flows, they fundamentally differ from our approach:

1. **Different Loss Functions**: Both FlowSVDD and OneFlow develop custom loss functions rather than using maximum likelihood estimation. FlowSVDD jointly learns a decision radius, breaking the direct connection to density estimation. OneFlow uses a custom loss based on Bernstein polynomial quantile estimates and volume minimizations.

2. **Different Theoretical Guarantees**: Their theoretical analyses focus solely on collapse prevention through bijectivity. Our argument relies on both bijectivity *and* the direct connection to maximum likelihood training—i.e., minimization of the reverse KL-divergence to a Gaussian in latent space—which provides provably non-degenerate optima (Section 4.2).

3. **Latent Space Alignment Analysis**: Neither work analyzes the monotonic alignment between data density and latent norm that we establish in Section 4.3. This property is crucial for understanding why USFs provide superior anomaly ranking and is not merely a consequence of avoiding collapse.

4. **Unified Perspective**: Our contribution lies in revealing that standard MLE training of USFs *already implements* an effective Deep SVDD variant with implicit regularization, unifying density estimation and one-class classification principles. This theoretical bridge has not been previously established in the literature.

### On Experimental Design and Future Work

We designed our experiments to isolate the causal effect of the USF substitution through controlled within-architecture comparisons. While this means absolute numbers may be lower than current leaderboards (as we did not perform model-specific tuning), it ensures that observed improvements are attributable to the architectural change rather than hyperparameter optimization artifacts. We are extending our evaluation to include additional datasets and modalities in the revised manuscript.

---

### Comment · Area_Chair_JxGc · 2025-11-28

Dear Reviewers,

Thank you for your valuable time and expertise in reviewing this paper.

The authors have now submitted their rebuttal. We would appreciate it if you could review their responses and assess whether your concerns have been addressed.

Best regards,

AC

---

### Author Response · Authors · 2025-12-03
**Final comment by the authors**

We thank the reviewers for their thoughtful feedback and constructive dialogue. In response to the consensus on broadening the empirical validation we have conducted comprehensive additional benchmark experiments on tabular data using ADBench. These new results directly compare our USFlows and VAE-USFlow against their non-US counterparts, as well as the most recent state-of-the-art Deep SVDD variants (DOHSC, and DO2HSC) and the most recent Flow–SVDD hybrid that also relies on a constant-Jacobian determinant flow mentioned by the reviewers (OneFlow, as FlowSVDD is a previous variant proposed by the same authors), across 47 classical tabular datasets. The experiments show that uniformly scaling flows, trained with standard maximum likelihood, perform on par with or outperform these methods while substantially improving performance stability, reinforcing that our theoretical bridge yields practical gains beyond image-based benchmarks.

This addition, detailed in the revised manuscript (Section 5.1), addresses the central request for validation against volume-preserving flows and other state-of-the-art baselines and demonstrates the generality of our approach. Together with the previously presented gains in stability and performance on MVTec AD and VisA for image anomaly detection, we now provide evidence that the USF principle is beneficial across data modalities.

Furthermore, we extended the ablation study with regard to the impact of the proposed log-determinant prior regularization, as well as the impact of changing the model backbone.

We have also corrected the minor technical issue in Proposition 1 and refined the narrative to further clarify our novelty relative to prior works like FlowSVDD and OneFlow: our core contribution is showing that *standard MLE training* of a USF inherently recovers a well-regularized Deep SVDD objective, guaranteeing density-aligned latent spaces without custom loss functions. Additionally, we added a discussion to directly relate the observed experimental results, especially the stability improvement, with our theoretical analysis.

We believe the strengthened empirical evaluation and clarified theoretical contributions fully address the concern of all three reviewers thoroughly. Especially, we kindly point out that the 3rd reviewer already explicitly indicated his willingness to raise their score given the continued relevancy of Deep SVDD, as well as the fix of the minor error in the formal derivation, and the strengthening of the empirical results through further data modalities and ablations.

Thank you again for your time and valuable insights, which we agree to that addressing them has significantly improved the quality of the paper.

---

### Meta-Review · Area_Chair_qTAr · 2026-01-07

**Summary:**

This paper proposes Uniformly Scaling Flows (USFs), a restricted class of normalizing flows where the Jacobian determinant is constant across the input space. We receive three reviews whose scores are 2, 4 and 6. Both the strengths and weaknesses are distinct. The strength is mainly on the novel and insightful monotonic density–norm alignment, while the weakness is on the experiment validation with only MVTec AD and VisA, making the performance gains hard to assess.  While no reviewer is willing to champion this paper, I tend to reject.

**Reviewer Concerns:**

The experiment validation with only MVTec AD and VisA are not addressed.

**Reviewer Scores:**

The scores are 2, 4 and 6. Reviewer U6Ni (4) have discussed with the authors but decided to keep the original score.

---

### Decision · Program_Chairs · 2026-01-26

Reject